# Mapping inhibitory sites on the RNA polymerase of the 1918 pandemic influenza virus using nanobodies

Jeremy R. Keown [1,8], Zihan Zhu[2,8], Loïc Carrique[1,8], Haitian Fan[2,8], Alexander P. Walker[2,6], Itziar Serna Martin[2,7], Els Pardon [3,4], Jan Steyaert [3,4], Ervin Fodor [2,9✉] & Jonathan M. Grimes[1,5,9✉]

Influenza A viruses cause seasonal epidemics and global pandemics, representing a considerable burden to healthcare systems. Central to the replication cycle of influenza viruses is the viral RNA-dependent RNA polymerase which transcribes and replicates the viral RNA genome. The polymerase undergoes conformational rearrangements and interacts with viral and host proteins to perform these functions. Here we determine the structure of the 1918 influenza virus polymerase in transcriptase and replicase conformations using cryo-electron microscopy (cryo-EM). We then structurally and functionally characterise the binding of single-domain nanobodies to the polymerase of the 1918 pandemic influenza virus. Combining these functional and structural data we identify five sites on the polymerase which are sensitive to inhibition by nanobodies. We propose that the binding of nanobodies at these sites either prevents the polymerase from assuming particular functional conformations or interactions with viral or host factors. The polymerase is highly conserved across the influenza A subtypes, suggesting these sites as effective targets for potential influenza antiviral development.

[1] Division of Structural Biology, Wellcome Centre for Human Genetics, University of Oxford, Oxford, UK. [2] Sir William Dunn School of Pathology, University of Oxford, Oxford, UK. [3] VIB-VUB Center for Structural Biology, VIB, Brussels, Belgium. [4] Structural Biology Brussels, Vrije Universiteit Brussel, Brussels, Belgium. [5] Harwell Science & Innovation Campus, Diamond Light Source Ltd, Didcot, UK. [6]Present address: School of Cellular and Molecular Medicine, Faculty of Life Sciences, University of Bristol, Bristol, UK. [7]Present address: Department of Virology, Faculty of Veterinary Sciences, Utrecht University, Utrecht, The Netherlands. [8]These authors contributed equally: Jeremy R. Keown, Zihan Zhu, Loïc Carrique, Haitian Fan. [9]These authors jointly supervised this work: Ervin Fodor, Jonathan M. Grimes. ✉email: ervin.fodor@path.ox.ac.uk; jonathan@strubi.ox.ac.uk

nfluenza A viruses are an important global health concern causing seasonal epidemics and, more rarely, global pandemics. The currently circulating influenza A viruses are thought to be the evolutionary progeny of the virus that caused the 1918–1919 global pandemic, which was responsible for between 50 and 100 million deaths worldwide. The 1918 influenza virus is thought to have jumped from waterfowl into humans and is considered the 'founder virus' that has contributed viral genome segments for all subsequent epidemic and pandemic strains. It remains unclear why this virus was so highly pathogenic, but several viral factors as well as secondary bacterial infections have been implicated in the high lethality of the virus[1].

The genome of influenza A virus is composed of eight different negative-sense RNA segments packaged in the form of viral ribonucleoprotein (vRNP) complexes. Each vRNP consists of oligomeric viral nucleoprotein (NP) and viral RNA (vRNA), which form a large loop arranged as a helical filament, with the viral polymerase bound to the 5′ and the 3′ ends of the vRNA[2]. The influenza virus RNA-dependent RNA polymerase is a heterotrimeric multifunctional machine composed of polymerase basic 1 (PB1), polymerase basic 2 (PB2) and polymerase acidic (PA) subunits, that catalyzes both primer-independent replication (vRNA to complementary RNA (cRNA) and cRNA to vRNA synthesis) and primer-dependent transcription (vRNA to mRNA synthesis) in the nucleus of infected cells. PB1 houses the polymerase active site, whereas PB2 and PA contain, respectively, cap-binding and endonuclease domains required for transcription initiation by cap-snatching[3–6]. High-resolution structures of the complete viral polymerase from a number of human, avian and bat influenza A viruses, as well as influenza B, C, and D viruses have been determined[7–11]. These structures have revealed that the polymerase is a highly flexible macromolecule that can assume a variety of distinct conformations, enabling the polymerase to perform both transcription and replication that require different initiation and termination mechanisms. The polymerase genes are among the most highly conserved genes of influenza viruses and as such the polymerase represents an ideal target for antivirals. Current influenza therapeutics target the neuraminidase and polymerase and inhibitors such as baloxavir (Xofluza) and favipiravir have been approved for use in a limited number of countries. However, mutations allow the virus to escape inhibition by these compounds and therefore further antivirals are required[12]. Recently, nanobodies, comprising the variable domain of single-chain camelid antibodies also referred to as VHH, have emerged as novel antiviral agents for both extracellular and intracellular targeting of viral proteins[13].

In this work, we determine structures of the polymerase from the 1918 pandemic influenza virus and identify sites on the surface of the polymerase that are sensitive to inhibition. Using single-particle analysis (SPA) cryo-EM and X-ray crystallography, we determine the binding location of 20 nanobodies on the RNA polymerase of the 1918 pandemic influenza virus. Functional analysis of the nanobodies shows that a subset of nanobodies strongly inhibits polymerase activity in vitro and in cell culture, and efficiently reduces virus growth. Combining functional and structural data we discover sites of vulnerability on the polymerase and link these to the inhibition of specific polymerase functions, identifying and validating targets for drug discovery.

## Results

**Structural characterisation of the 1918 pandemic influenza virus polymerase.** We co-expressed the three subunits of the RNA polymerase from influenza A/Brevig Mission/1/1918 (H1N1) virus in insect cells and purified the heterotrimeric complex. In vitro transcription assays showed that the recombinant polymerase was active (Supplementary Fig. 1). Polymerase with added capped RNA primer and vRNA promoter, comprising the 5′ and 3′ terminal sequences of vRNA, was used for EM grid preparation. We observed monomeric polymerase heterotrimers as well as both dimeric and tetrameric assemblies (Supplementary Fig. 2). The dimeric polymerase species were highly similar to the previously observed dimer implicated in cRNA to vRNA synthesis[7]. However, severe preferential orientation prohibited accurate 3D reconstruction of both dimeric and tetrameric forms. After 2D and initial 3D classification of the monomeric particles a consensus refinement of the particles yielded a high-quality map. Further 3D classification of this map yielded four maps into which models were built, resulting in four distinct polymerase structures. In all four structures the core of the polymerase composed of PB1, the C-terminal domain of PA and the N-terminal one third of PB2 was fully resolved, while the flexible domains including the N-terminal endonuclease domain of PA and the C-terminal two thirds of PB2 showed different arrangements or remained unresolved. To aid in map interpretation the programme deepEMhancer was used for low abundance conformations (Supplementary Fig. 3)[14]. Both the 5′ and 3′ vRNA promoters, but no capped RNA primer, was observed in all structures. The 5′ promoter was fully ordered forming the hook structure bound by PB1 and PA as observed previously[11]. Only eight bases at the 5′ end of the 3′ promoter were observed, four of which form a duplex with the 5′ promoter while the remaining four are oriented towards the polymerase active site. The seven 3′ terminal unresolved bases of the 3′ promoter strand have likely entered the active site but remain unresolved.

The most complete structure (class 1), representing ~15% of the particles, shows an almost completely ordered polymerase heterotrimer with only the C-terminal nuclear localisation signal (NLS) domain of PB2 remaining unresolved (Fig. 1a). The overall conformation of the polymerase is similar to the previously observed conformations implicated in transcription although with some differences in the exact location and orientation of the C-terminal domains of PB2, comprising the mid, cap-binding, linker (which together with mid forms a rigid-body domain midlink), 627 and NLS domains[7,15,16]. Specifically, the cap-binding domain is rotated by ~90° relative to the polymerase core with the cap-binding site facing away from the product exit/primer entry channel (Supplementary Fig. 4a). This conformation of the cap-binding domain is stabilised through a correlated rotation of the 627 domain by 26°. The co-rotation of these domains leads to a new interaction between the 424 loop of the cap-binding domain and residues 585–588, 624 and 625 of the 627 domain, instead of the previously observed interaction of the 424 loop with PB1 residues[7,15,17]. However, the overall fold of the 627-domain remains the same as observed in previous structures.

As shown previously, the PB2 cap-binding domain can rotate during transcription[11] and here we identify another stable conformation. This conformational flexibility allows the cap-binding domain to probe the environment for capped RNA and capture the cap, orient the uncleaved capped RNA into the PA endonuclease domain and then insert the cleaved capped primer into the polymerase active site through the product exit/primer entry channel. We propose two possible roles for the observed conformation. This cap-binding domain position could represent a conformation where the polymerase is probing the environment, waiting to capture an uncleaved capped RNA still associated with host RNA polymerase II[5]. Alternatively, this may be an intermediate position that occurs after cleavage of the capped RNA but before the primer enters the polymerase active site. The cap-binding domain moving away from the polymerase core may help it to accommodate primers of variable length,

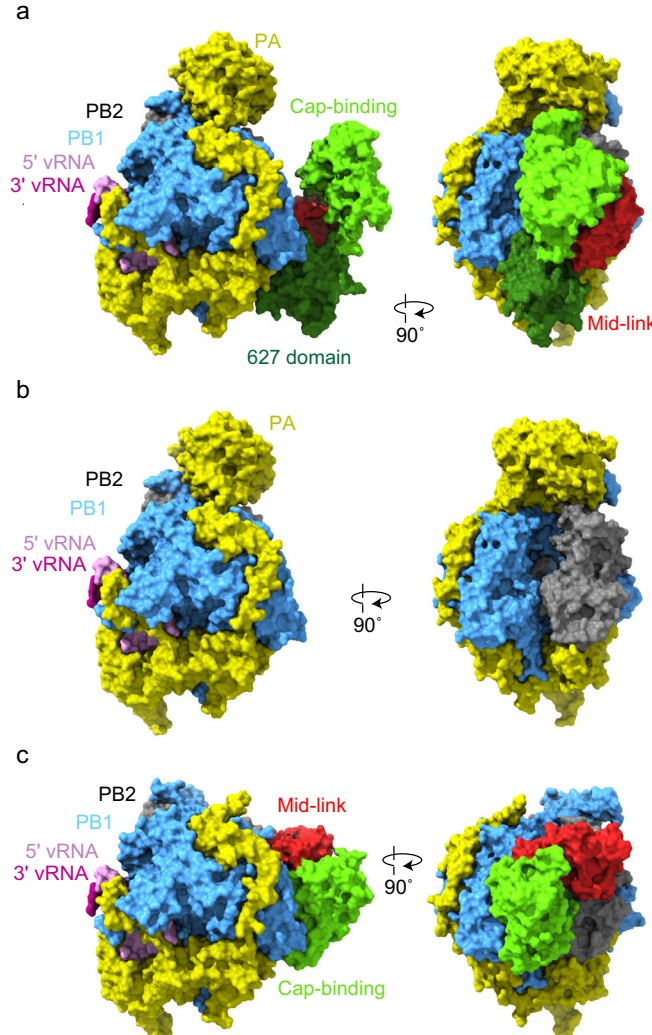

**Fig. 1 Cryo-EM structures of the 1918 polymerase in three conformations.**
Monomeric conformations of the 1918 polymerase in full transcriptase
(class 1) (**a**), partial transcriptase (class 2a) (**b**), or replicase conformations
(class 3) (**c**).

reported to range between 8 and 15 nucleotides[18–20], and orient
their 3′ end into the active site.

The class 2a and class 2b structures, representing ~78% of the
particles, are similar to the class 1 structure; however, the flexible
C-terminal domains of PB2 remain entirely unresolved (Fig. 1b).
The two structures (class 2a and class 2b) differ in the orientation
of the PA endonuclease and the ordering of the priming loop, a
hairpin that protrudes from the PB1 thumb subdomain into the
polymerase active site and is involved in the positioning of the
template and/or initiating nucleotide in the polymerase active
site[7,15,21]. For class 2a the location and orientation of the PA
endonuclease relative to the polymerase core is identical to that in
the class 1 structure and very similar to that previously observed
in the transcriptase conformations of the polymerase[7,15]. In the
class 2a structure, as in the class 1 structure, the priming loop is
fully ordered despite no template RNA being resolved in the
active site. In the class 2b structure, the PA endonuclease,
together with the C-terminal helices of PB1 and the N-terminal
helix of PB2 with which it forms a rigid body, is rotated by 20°
relative to the core of the polymerase (Supplementary Fig. 4b).
This rotation leads to new interactions between the PA
endonuclease and the polymerase core mediated through PA

Lys22 interacting with PB1 Thr156, Glu159, and Ser160
(Supplementary Fig. 4b). Interestingly, this rotation correlates
with disordering of the priming loop.

The last structure (class 3), representing ~7% of particles,
shows a polymerase with a fully ordered core, including the
priming loop, and the PB2 cap-binding and mid-link domains,
but lacks the PA endonuclease and the PB2 627 and NLS domains
(Fig. 1c). The resolved PB2 domains are arranged in a
conformation similar to that observed in crystal structures of
the apo human H3N2 and avian H5N1 influenza A virus and
influenza C virus polymerases as well as cRNA-bound influenza B
virus polymerase and, more recently, the cryo-EM structure of the
influenza C virus polymerase dimer in complex with host acidic
nuclear phosphoprotein 32A (ANP32A), described as a replicase
(Supplementary Fig. 4c)[7,8,17,22]. It is tempting to speculate that, in
solution, in the absence of ANP32A and polymerase dimer
formation there is nothing against which the PB2 627 domain
could pack and so it remains flexible.

Taken together, our characterisation of the polymerase of the
1918 pandemic influenza virus has yielded a number of important
observations. The polymerase is highly dynamic with multiple
oligomers and conformations present in a single sample. We
observe the flexible PB2 C-terminal domains in conformations
described for both the transcriptase and replicase conformations,
while most molecules show no ordered PB2 C-terminal domains.
No vRNA template is observed in the polymerase active site;
however, we are able to resolve the priming loop in most
structures. These data suggest the ordering of the priming loop is
uncoupled from the ordering of RNA in the polymerase active
site and is perhaps related to the position of the PA endonuclease
domain. We were able to resolve the flexible C-terminal domains
of PB2 in a new arrangement, similar to that described for the
transcriptase[7,15]. Specifically, we resolved the PB2 cap-binding
domain in a new orientation that we propose is important for
capturing an uncleaved capped RNA or orienting the cleaved
capped RNA primer into the active site. Interestingly, no capped
RNA was observed although the reasons for this remain unclear.
We envisage that these structural insights hold true for all
influenza A virus polymerases as there is a very high degree of
sequence homology among the A strains (Supplementary
Fig. 5a–c), although we cannot exclude the possibility that some
of these conformations are due to unknown characteristics of the
1918 influenza polymerase.

**Structural characterisation of the 1918 influenza virus poly-
merase in complex with nanobodies.** A panel of 24 nanobodies
(Nb8189-Nb8212) was generated against recombinantly expres-
sed and purified polymerase of an H5N1 influenza A virus as
described previously[7]. Using cryo-EM we determined the struc-
ture of 17 polymerase-nanobody complexes at resolutions
between 4.5 and 6.7 Å; this is sufficient resolution to accurately
position each nanobody bound to the polymerase heterotrimer
(Supplementary Fig. 6, Supplementary Table 1). In addition to the
core of the polymerase which was resolved in all structures, the
PA endonuclease was resolved in the majority of structures
(excluding Nb8189, Nb8190, Nb8192, Nb8196, Nb8202), while
the C-terminal domains of PB2 remained unresolved in all
structures. The nanobody binding sites were spread across the
surface of the polymerase with related nanobodies binding at the
same site (Supplementary Fig. 7a, b).

The most populous binding site, termed site 1, is on the PA
endonuclease with five nanobodies (Nb8198, Nb8199, Nb8200,
Nb8203, Nb8209) that bind at an interface formed by the short
helix α1 (residues 1–10) and the long helix α6 (170–185) (Fig. 2a).
On the PA linker (site 2), that links the N-terminal PA

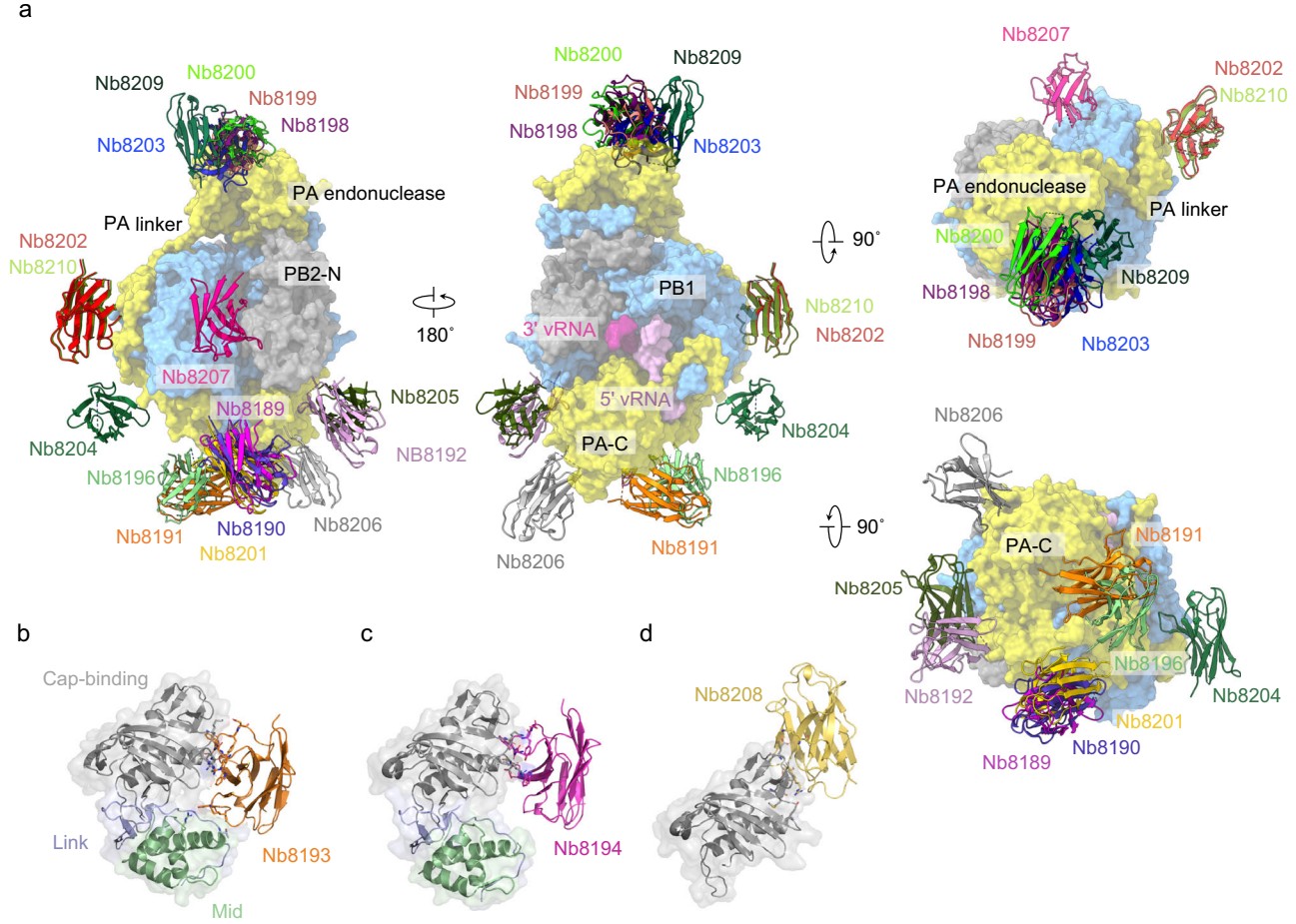

**Fig. 2 Nanobody binding sites of the 1918 polymerase determined by cryo-EM and X-ray crystallography. a** Composite model showing the binding position of each nanobody on the surface of the 1918 polymerase determined using cryo-EM. C-terminal domain of PA, PA-C; N-terminal one third of PB2, PB2-N. **b-d** Crystal structures of PB2 cap-mid-link and nanobody complexes.

endonuclease domain and the PA C-terminal domain and wraps around PB1, we observe three nanobodies; Nb8202 and Nb8210 bind identically at residues 220–240 and Nb8204 binds at residues 250–265. Nb8207 binds to a unique position, termed site 3, at the product exit channel contacting both PB1 and the N-terminal region of PB2. The PA C-terminal domain is the binding site for eight nanobodies which cluster into four groups (sites 4a–d) across the surface of the domain. Nb8192 and Nb8205 bind to PA residues 426–445 (site 4a) with Nb8205 also making a minor contact with the PB1. Three nanobodies (Nb8189, Nb8190, Nb8201) bind at an interface formed by PA residues 615–630 and PB1 residues 10–15 (site 4b). Nb8191 binds to the PB1 N-terminus and PA residues 405–420, 531–534, 551–557, and 624–626 (site 4c). Nb8196 somewhat overlaps with Nb8191, binding to PA residues 262–268, 624–626 and 705–712. Finally, Nb8206 binds uniquely to PA residues 309–321 and 555–558 (site 4d).

Five nanobodies, Nb8193, Nb8194, Nb8195, Nb8197, and Nb8208, were unresolved by cryo-EM, and as the C-terminal domains of PB2 are disordered in our datasets, we suspected that these nanobodies may bind to this region. To test this hypothesis, we purified proteins comprising either the cap-binding and mid-link domains (cap-mid-link, residues 250–538) or the 627 and NLS domains (627-NLS, residues 538–759). Four of the nanobodies bound to cap-mid-link causing a peak shift in size exclusion chromatography; only Nb8195 did not bind (Supplementary Fig. 7c). Crystallisation of the cap-mid-link-nanobody complexes yielded crystals that diffracted to resolutions of 1.7 Å

for the Nb8193 complex and 1.9 Å for the Nb8194 complex (Fig. 2b–d, Supplementary Table 2). Nanobodies Nb8193 and Nb8194 form site 5a, each interacting with the cap-binding domain residues Arg389, Asp390, and Arg482. Nb8193 forms additional contacts with Arg264 from the mid domain and Thr524 from the linker region. We also determined the structure of Nb8208 bound to the cap-binding domain at a resolution of 3.1 Å, a product of in-drop proteolysis from crystallisation screens set up with the complete cap-mid-link construct. Nb8208 binds to site 5b interacting with residues Glu391, Asp466, The468, and Glu472 (Fig. 2d). The cap-mid-link-Nb8197 complex failed to produce diffraction quality crystals but, based on the similarity of its complementarity-determining regions (CDRs), Nb8197 likely binds at a similar position as Nb8193 and Nb8194. We have not determined the structures of Nb8211 and Nb8212 which, based on their similarity to Nb8200 and Nb8198 (Supplementary Fig. 7a, b), respectively, are expected to bind to the PA endonuclease.

**Nanobodies inhibit polymerase activity in vitro and viral growth in cell culture.** To assess the inhibitory effect of nanobodies on polymerase activity, we performed an in vitro polymerase activity assay using recombinant RNP complexes derived from the 1918 pandemic influenza virus. β-globin mRNA was added to the vRNPs to facilitate primer-dependent transcription, in the presence of an excess of purified nanobody. These assays showed that a subset of nanobodies significantly inhibit

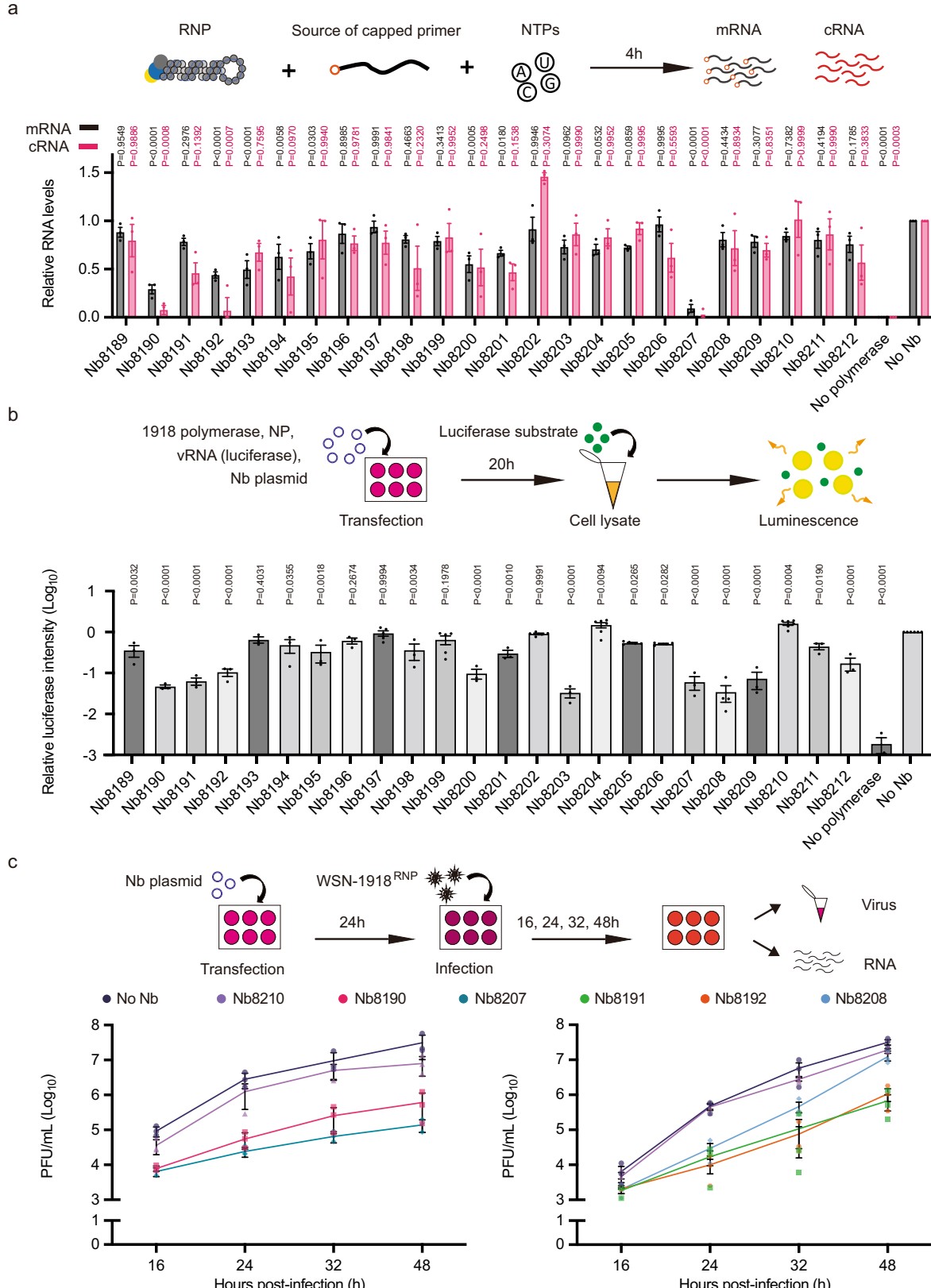

transcription and/or vRNA to cRNA replication with Nb8207 being particularly potent (Fig. 3a, Supplementary Fig. 8a). Interestingly, we observe different potencies for nanobodies that bind at similar locations, for example Nb8189 and Nb8190 which bind the same site on the C-terminal domain of PA; this is likely due to differences in the affinities of each nanobody for a given

site. To investigate the effect of nanobodies on polymerase activity in cellular environment, we used a minireplicon luciferase-reporter assay, in which each nanobody was expressed along with polymerase subunits and NP from the 1918 pandemic influenza virus and a negative-sense vRNA encoding luciferase (Fig. 3b). This assay revealed that most nanobodies, including the

**Fig. 3 Effect of nanobodies on 1918 polymerase function and virus growth. a** Effect of nanobodies on the activity of recombinant 1918 RNP in the presence of a source of capped primer. Data are mean ± s.e.m. $n = 3$ independent RNP purifications and reactions. Ordinary one-way ANOVA was used to compare mRNA and cRNA levels in the absence (no Nanobody) and presence of the indicated nanobody. $P < 0.05$ is considered significant. **b** Effect of nanobodies on 1918 influenza polymerase activity using a luciferase-reporter minireplicon assay. Data are mean ± s.e.m. $n = 3$ independent transfections with $n = 2$ technical replicates. Ordinary one-way ANOVA was used to compare the relative luciferase intensity in the presence and absence of nanobodies. $P < 0.05$ is considered significant. **c** Effect of two sets of nanobodies on the replication of a reassortant A/WSN/1933 virus encoding PB1, PB2, PA and NP of A/Brevig Mission/1/1918 (WSN-1918$^{RNP}$) virus in HEK293T cells. Data are mean ± s.e.m. $n = 3$ independent transfections and infections for each set of nanobodies. PFU plaque-forming units. Ordinary one-way ANOVA was used to compare the viral titres in the presence and absence of nanobodies at a given time point. $P$ values are as follows: for Nb8190 $P = 0.0007$, $P = 0.0067$, $P = 0.0331$ and $P = 0.0199$, for Nb8207 $P = 0.0006$, $P = 0.0063$, $P = 0.0301$ and $P = 0.0185$, for Nb8210 $P = 0.0084$, $P = 0.0925$, $P = 0.3444$ and $P = 0.0696$, for Nb8191 $P = 0.1030$, $P = 0.0009$, $P = 0.0141$ and $P < 0.0001$, for Nb8192 $P = 0.1227$, $P = 0.0008$, $P = 0.0136$ and $P = <0.0001$, for Nb8208 $P = 0.1179$, $P = 0.0011$, $P = 0.0205$ and $P = 0.0025$, and for Nb8210 $P = 0.7246$, $P = 0.9865$, $P = 0.2269$ and $P = 0.0340$, at 16, 24, 32 and 48 h post infection, respectively. $P < 0.05$ is considered significant. For quantification of viral RNA levels during the course of infection, see Supplementary Fig. 8b. Source data are provided as a Source Data file.

ones identified as inhibitory in the in vitro assay, show strong potency in cells. Some nanobodies, such as those binding to the PA endonuclease domain which showed no or minor inhibition in the in vitro assay were strongly inhibitory in the minireplicon assay, suggesting that this surface of the endonuclease is involved in an important function that is not required for in vitro activity but is essential in the more complex minireplicon assay.

Based on data from the in vitro and minireplicon assays, we selected a subset of inhibitory nanobodies that bind at different sites and carried out viral growth assays in cells expressing the selected nanobody prior to infection with a reassortant influenza A/WSN/33 virus encoding polymerase and NP of the 1918 pandemic influenza virus (Fig. 3c). Nb8190, Nb8191, Nb8192, and Nb8207 caused a significant reduction in viral titre at most time points tested. Nb8208 showed a strong inhibition at the early time points but viral titres were not reduced at 48 h post infection. Nb8210, used as a negative control, had only a minor or no effect on viral growth as reported previously[7]. Viral RNA quantification was carried out at each time point and showed a decrease in mRNA, cRNA, and vRNA levels that match the decrease in viral titres (Supplementary Fig. 8b). Nanobody expression was verified by western blot and had no effect on cell viability (Supplementary Fig. 9a, b).

Next, we evaluated the inhibitory mechanisms of these nanobodies. To assess the effect of nanobodies on viral transcription we first addressed whether they have an effect on primary transcription, i.e. the production of viral mRNA from the incoming vRNPs. To assay the effect of nanobodies on this process we performed a primary transcription assay by infecting cells that had been pre-transfected with plasmids to express each of the selected nanobodies. Protein synthesis was blocked at the onset of infection through addition of cycloheximide, which prevented cRNA accumulation and vRNA synthesis, isolating the viral transcription process from replication. Nb8190, Nb8191, Nb8192 and Nb8207 were shown to significantly inhibit viral mRNA synthesis (Fig. 4a). During infection the influenza virus polymerase must interact with the C-terminal domain (CTD) of the large subunit of host RNA polymerase II (Pol II) to facilitate access to 5′ capped RNA primers snatched from nascent Pol II transcripts that are used by the viral polymerase to prime viral mRNA synthesis[23–25]. To probe the effect of nanobodies on this interaction, the binding of purified recombinant 1918 polymerase to a Pol II CTD mimic peptide consisting of four heptad repeats, phosphorylated on serine-5 of the heptad repeats, was measured in an in vitro binding assay. Nb8191 and Nb8192 were found to significantly reduce the binding of the polymerase to the CTD peptide (Fig. 4b).

To assess the effect of each nanobody on the first step of viral genome replication, vRNA to cRNA synthesis, we carried out a cRNA stabilisation assay[26]. In this assay, prior to infection and

cycloheximide treatment, the cells are transfected to express influenza virus NP and polymerase subunits including a catalytically inactive PB1 subunit (PB1a). Under these conditions cRNPs are produced which are inactive due to the incorporation of the inactive PB1 subunit. Quantifying the cRNA from these infections shows that all the selected nanobodies significantly reduced cRNA accumulation (Fig. 4c). To assay for the second step of replication, cRNA to vRNA synthesis, we replaced the catalytically inactive PB1 with wild-type to allow replication to proceed and produce progeny vRNA. All selected inhibitory nanobodies dramatically decreased vRNA accumulation and caused a reduction in secondary mRNA production (Fig. 4d).

Finally, to address whether the observed inhibitory effects can be generalised across different influenza A virus subtypes, we performed minireplicon luciferase-reporter assays using polymerase subunits and NP from influenza A viruses belonging to H3N2, H5N1 or H1N1 subtypes. These results demonstrate that the inhibitory effect of the nanobodies is conserved across different influenza A virus subtypes (Supplementary Fig. 9c).

## Discussion

In this study we have solved the structure of the RNA polymerase of the 1918 pandemic influenza virus in four distinct conformations. These conformations show features common with previously determined structures implicated in either transcription or replication. In a conformation characteristic of a transcriptase we observed a novel arrangement of the PB2 cap-binding domain. This finding is in agreement with the idea that this domain is extremely flexible, which allows it to sample the environment for nascent host capped RNA, bind to it and then orient it towards the PA endonuclease for cleavage, followed by positioning of the short capped RNA product into the polymerase active site for transcription initiation. We also observed a correlation between the ordering of the priming loop in the active site and the position of the PA endonuclease, suggesting crosstalk between the two. However, the significance of this observation remains unclear.

In our attempt to map inhibitory sites on the polymerase of the 1918 pandemic influenza virus we have determined the structure of 20 nanobodies bound to the polymerase heterotrimer or a fragment of the PB2 subunit. A combination of in vitro and cell-based polymerase assays and virus growth inhibition assays led to the identification of multiple sites on the polymerase surface that are sensitive to inhibition by nanobodies (Fig. 5a, b). In order to transcribe viral genes and replicate the viral RNA genome the influenza virus RNA polymerase must interact with viral and host factors and adopt at least three distinct conformations to act as a transcriptase, replicase, or encapsidase[4]. We speculate that binding of nanobodies most likely inhibits polymerase function by locking it in a particular conformation, preventing it from

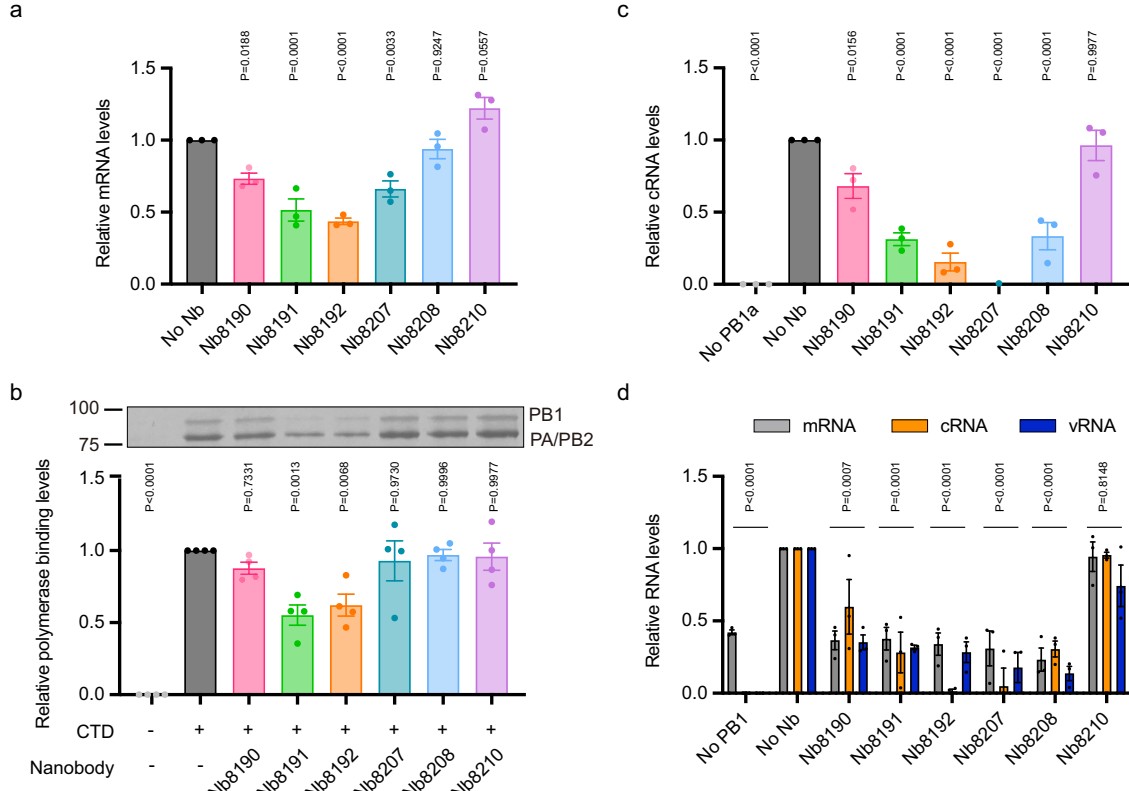

**Fig. 4 Inhibitory mechanisms of nanobodies. a** Effect of nanobodies on primary transcription by a reassortant A/WSN/1933 virus polymerase and NP of A/Brevig Mission/1/1918 virus in HEK293T cells transfected to express the indicated nanobodies. Data are mean ± s.e.m. $n = 3$ independent transfections and infections. Ordinary one-way ANOVA was used to compare the relative mRNA levels in the presence and absence of nanobodies. $P < 0.05$ is considered significant. **b** Effect of nanobodies on binding of purified recombinant 1918 pandemic influenza virus polymerase to serine-5-phosphorylated Pol II CTD mimic peptide. Top, a representative silver-stained gel. Bottom, quantification of gels from $n = 4$ independent binding assays. Data are mean ± s.e.m. Ordinary one-way ANOVA was used to compare the relative polymerase binding in the presence and absence of nanobodies. $P < 0.05$ is considered significant. **c, d** Effect of nanobodies on cRNA levels produced by a reassortant A/WSN/1933 virus encoding polymerase and NP of A/Brevig Mission/1/1918 virus in HEK293T cells transfected to express the indicated nanobodies and influenza virus NP and polymerase subunits, including a polymerase active site mutant PB1 (PB1a) (**c**) or wild-type PB1 (**d**). Plasmid to express PB1a (no PB1a) (**c**) or PB1 (no PB1) (**d**) was omitted as negative control. Data are mean ± s.e.m. $n = 3$ independent transfections and infections. Ordinary one-way ANOVA was used to compare the relative cRNA levels (**c**) or RNA levels (mRNA, cRNA and vRNA combined) (**d**) in the presence and absence of nanobodies. $P < 0.05$ is considered significant. Source data are provided as a Source Data file.

assuming a particular functionally important conformation or inhibits interactions with viral or host factors.

The site bound by Nb8207, one of the most potent nanobodies identified in our study, is located at the product exit channel, formed by PB1 and the N-terminal region of PB2. Binding of a nanobody at this site inhibits RNA production in both transcription and replication without affecting other polymerase activities such as CTD binding (Figs. 4a–d, 5a, b). We speculate that nanobody binding could inhibit RNA exit by blocking the product exit channel or might prevent the polymerase from assuming an active conformation as binding of a nanobody at this site is incompatible with the packing of the C-terminal domains of PB2 in both transcriptase and replicase conformations.

The site bound by Nb8208 is located on the cap-binding domain of PB2. This region of the PB2 cap-binding domain packs against the PB1 palm subdomain in the encapsidase conformation of the polymerase as observed in the dimeric polymerase bound by the host protein ANP32A (Fig. 5a, b)[22]. Nanobody binding at this site severely affected both steps of replication but had no effect on primary transcription, Pol II CTD binding and polymerase activity in vitro in agreement with the idea that binding of Nb8208 affects formation of the ANP32A-bound dimeric polymerase.

We have identified several inhibitory sites on the C-terminal domain of PA. The site bound by Nb8190 corresponds to a site against which the 627 domain of PB2 is packed in the transcriptase conformation, potentially preventing the polymerase from assuming the transcriptase conformation. Accordingly, Nb8190 inhibited transcription although it had no effect on the binding of the polymerase to Pol II CTD (Fig. 4a). The site bound by Nb8191 overlaps with the binding site of the Pol II CTD[23,24] and, indeed, this nanobody was found to inhibit the binding of the viral polymerase to the CTD of Pol II and, consequently, inhibit transcription (Fig. 4a, b). The site bound by Nb8192 overlaps with the polymerase dimer interface that has been identified as important for cRNA to vRNA replication[7]. Binding at this site would therefore prevent polymerase dimerisation and, in agreement, Nb8192 was found to inhibit RNA replication (Fig. 4c, d), as demonstrated previously for Nb8205[7] which has a binding site overlapping with that of Nb8192. Although the site bound by Nb8192 has not been shown to contribute to Pol II CTD binding our data suggest that it might play a role as Nb8192 also inhibited Pol II CTD binding and transcription. Overall, our data reinforce previous findings that the PA C-terminal domain plays a central role in multiple functions during RNA synthesis including Pol II CTD binding during transcription and

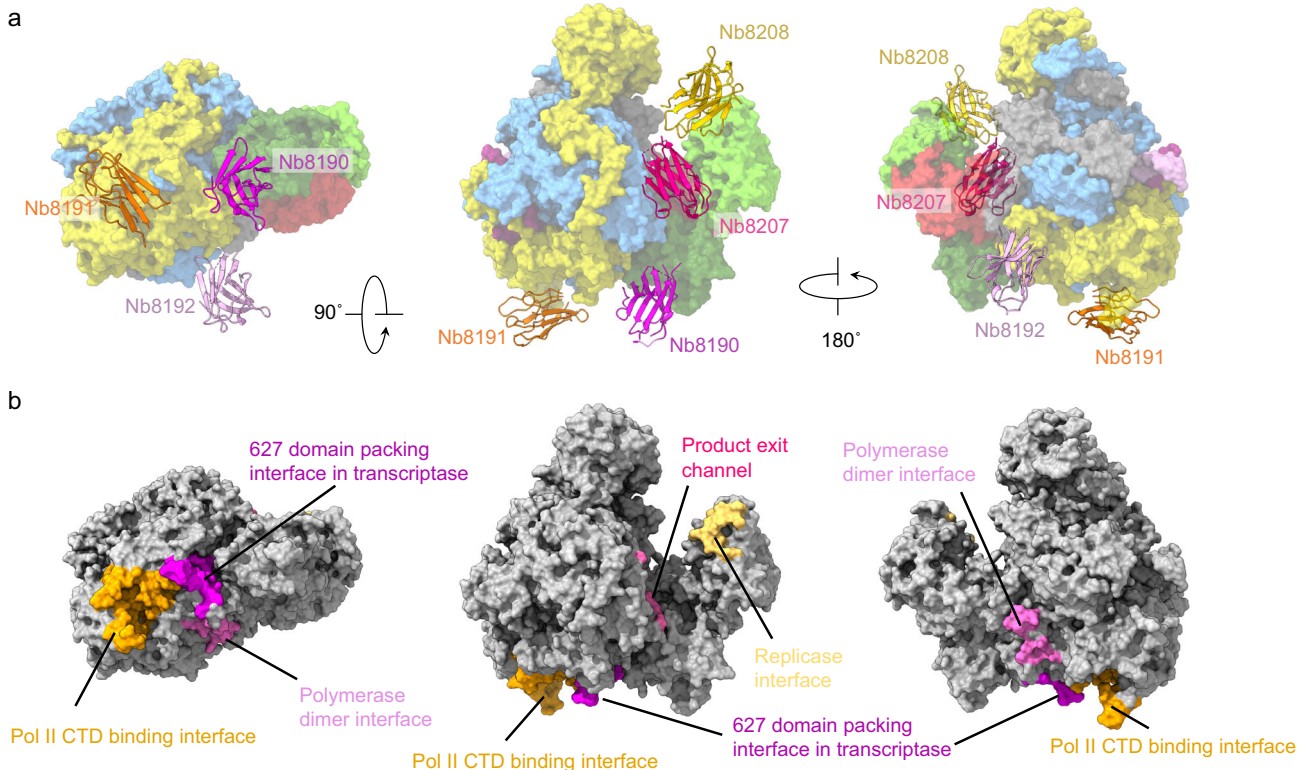

**Fig. 5 Inhibitory sites on the 1918 polymerase. a** Binding position of inhibitory nanobodies mapped on to the transcriptase conformation of the polymerase (PDB ID: 7NHX). Note that the binding of Nb8190 and Nb8207 is for illustration purposes only; the binding of these nanobodies is incompatible with the transcriptase conformation shown. Subunits and domains are coloured as follows: PA (yellow), PB1 (blue), PB2 N-terminal domain (grey), cap-binding domain (bright green), mid-link (red), 627 domain/NLS (dark green). 5′ and 3′ vRNA promoters are coloured purple and pink, respectively. **b** Location of each nanobody interaction site annotated with biological function. C-terminal domain of host RNA polymerase II, Pol II CTD.

providing a polymerase dimerisation interface during replication (Fig. 4a–d).

In summary, we have solved the structure of the RNA polymerase of the 1918 pandemic influenza virus. We have also identified sites on the polymerase surface that are sensitive to inhibition by nanobodies and proposed inhibitory mechanisms. These inhibitory sites are highly conserved across the polymerases of influenza A virus strains (Supplementary Figs. 5a–c, 9c) and as such present exciting therapeutic targets for small molecule compounds or inhibitory peptides.

## Methods

No statistical methods were used to predetermine sample size. The experiments were not randomised and investigators were not blinded to allocation during experiments and outcome assessment.

**Cells**. Human embryonic kidney 293T (HEK293T), Madin-Darby Bovine Kidney (MDBK), and Sf9 insect cells were sourced from the Cell Bank of the Sir William Dunn School of Pathology. HEK293T cells were maintained in Dulbecco's modified Eagle medium (DMEM) + 10% Fetal Calf Serum (FCS) and Sf9 cells were maintained in Sf-900 II serum-free medium. Cell lines have not been authenticated but tested negative for mycoplasma contamination.

**Protein expression and purification**. The three polymerase genes from influenza A/Brevig Mission/1/1918 (H1N1) virus were codon optimised for insect cells and synthesised (Synbio Technologies). Genes were then cloned into the Multibac system with protein expression and purification carried out as previously described[7]. The nanobodies were generated and purified as previously described[7,27]. In brief, plasmids encoding C-terminally His-tagged nanobodies were transformed into Escherichia coli WK6, grown in LB supplemented with 0.1% glucose (w/v), 1 mM MgCl$_2$, and 100 mg/ml ampicillin at 37 °C to OD$_{600}$ = 0.7, induced overnight with 1 mM isopropylthiogalactoside (IPTG) at 28 °C and collected by centrifugation (3500 × $g$, 15 min). Nanobodies were released from the periplasm by osmotic shock. Soluble nanobodies were separated from the proto-plasts by centrifugation (4000 × $g$, 30 min), and recovered from the clarified

supernatant by binding to Ni-NTA resin (Qiagen). Nanobodies were next eluted from Ni-NTA resin by applying 500 mM imidazole and concentrated by centrifugation using Amicon Ultra Filters (Sigma, 10 kDa cut-off). Nanobodies were buffer exchanged into 25 mM HEPES-NaOH pH 7.5, 150 mM NaCl for storage at −20 °C. Sequence comparison of the nanobodies was carried out using Clustal W[28]. The cap-mid-link construct (residues 247–536 of PB2) from influenza A/duck/Fujian/01/2002 (H5N1) was expressed and purified as previously described[29].

**Cryo-EM sample preparation**. To determine the structure of the polymerase an approach similar to that described by Kouba et al was utilised[16]. Briefly, polymerase was mixed with a 1.2 molar excess of vRNA promoters (5′ vRNA 5′-AGUA-GAAACAAGGCC-3′, 3′ vRNA 5′-GGCCUGCUUUUUGCU*AUU*-3′ with a 3 nucleotide long extension at the 3′ end (italics)) and a 50 molar excess of Baloxavir acid (Aobious) and left on ice for 10 min. Subsequently a 1.5 molar excess of synthetic capped mRNA primer (m$^7$GpppAm-AUCUAUAAUAG) and MgCl$_2$ to a final concentration of 5 mM were added. To allow limited extension of the product, nucleotides were added to a final concentration of 1 mM CTP, and 1 mM ApCpp (adenosine-5′-[(α,β)-methyleno]triphosphate). The reaction mixture was incubated at 20 °C overnight before being applied to a Superdex 200 Increase 10/300 GL in a buffer containing 20 mM HEPES-NaOH, pH 7.5, 500 mM NaCl. The eluted sample was concentrated to 0.9 mg ml$^{-1}$ and before grid preparation the sample was diluted 1:3 with HEPES-NaOH, pH 7.5, 37.5 mM NaSCN, 0.0075% (v/v) Tween20 to give a final protein concentration of 0.25–0.3 mg ml$^{-1}$. A volume of 3.5 µl was applied to glow discharged Quantifoil Holey Carbon (R2/1, 200 mesh, either gold or copper), blotted for 3.0–3.5 s and plunge frozen in liquid ethane. Grids were prepared using a Vitrobot mark IV (FEI) at 100% humidity. To prepare polymerase-vRNA-nanobody complex, first polymerase was mixed with a 1.2 molar excess of vRNA promoters (5′ vRNA 5′-pAGUAGAAACAAGGCC-3′, 3′ vRNA 5′-GGCCUGCUUUUUGCU-3′) and applied to a Superdex 200 Increase 10/300 GL column in a buffer containing 20 mM HEPES-NaOH, pH 7.5, 500 mM NaCl. Then a 1.5 molar excess of purified nanobody was added to vRNA-bound polymerase concentrated to 0.9 mg ml$^{-1}$. Grids were prepared as above.

**Cryo-EM image collection and processing**. For the initial structure determination of the polymerase, data was collected on a 300 kV Titan Krios equipped with a K2 Summit (Gatan) camera and a GIF Quantum energy filer at the Oxford Particle Imaging Centre. The data was collected using SERIALEM[30]. Data was processed

on-the-fly using cryoSPARC-Live V2.15[31]. Data were motion corrected and the CTF estimated using patch motion correction and patch CTF estimation, respectively (Supplementary Fig. 2). Particles taken from a subset of the dataset, picked using the cryoSPARC blob picker, were used to generate templates for complete picking using the cryoSPARC template picker. To remove bad particles, 2D classes were generated before particles from good classes were subjected to ab initio model generation followed by heterogeneous refinement. Particles from one class were then refined with non-uniform refinement and per particle CTF refinement to give a consensus refinement to 2.79 Å. Further heterogeneous refinement using eight copies of the consensus model enabled the discrimination of three discrete conformations within the dataset. The conformations with resolved polymerase core and PA endonuclease were further classified using 3D-variability analysis and models showing similarly ordered priming loop and endonuclease density were combined (Class 2a/b). These were all subsequently refined using a final non-uniform refinement[32]. To assist with the interpretation of maps for the other two conformations the deepEMhancer programme[14] was used to locally sharpen the complete transcriptase (Class 1) and partial replicase (Class 3) conformations. The wideTarget model provided the most easily interpretable maps. A comparison of the unsharpened and deepEMhancer maps is shown in Supplementary Fig. 3.

For model building the structure of the A/NT/60/68 influenza virus polymerase (PDB ID: 6RR7) was used as an initial model that was first changed to the polymerase sequence of the A/Brevig Mission/1/1918 influenza virus then rigid-body fit into the map with ChimeraX V1.1[33]. Iterative cycles of COOT[34] and real-space refinement in PHENIX[35] were used to improve model geometry and the fit to the map. Model geometry was monitored using MolProbity[36]. For the complete transcriptase and partial replicase conformations, the model of the polymerase core with the resolved PA endonuclease and priming loop PL was first positioned using ChimeraX V1.14[37] before the PB2 C-terminal domains were generated and manually placed into the appropriate density. These models were then rigid-body real-space refined in PHENIX. Manual adjustment of the C-terminal PB2 domains in COOT was used to achieve domain connectivity and make gross changes to the model. Due to the somewhat poorly resolved nature of these regions only restrained real-space refinement was carried out on these areas.

The polymerase-vRNA-nanobody complex data were collected on a 300 kV Titan Krios microscope with either a K2 Summit (Gatan) and GIF Quantum energy filter or Falcon 3 detector or on a 200 kV Glacios (Thermo Fisher Scientific) microscope with a Falcon 3 detector at the Oxford Particle Imaging Centre. Automated data collection was carried out using EPU versions 2.3–2.5. Sample specific cryo-EM data collection parameters have been summarised in Supplementary Fig. 2. Data processing was carried out within cryoSPARC versions V2.12–3.0. Briefly, raw movies were first motion corrected, before the contrast transfer function was estimated by Gctf v1.06[38] and micrographs of poor quality were removed. Particles were subsequently picked by the blob picker and classified by 2D classification to provide templates for template based picking. Iterative 2D classification produced a pool of particles from which three initial models were generated, with the classes showing extra density further refined with a non-uniform refinement. For some datasets, particles were further classified with 3D-variability analysis in cryoSPARC V2.13–2.15. As the maps were not of sufficient quality to build the complementarity-determining regions (CDRs) ab initio, a model for each nanobody was generated using SWISSMODEL[39], and the CDR loops removed. This approach was taken as modelling of CDRs is highly inaccurate and this approach avoids possible misinterpretation of our models. To generate models, the resolved polymerase core with PA endonuclease and priming loop conformation (Class 2b) and the nanobody structure was fit in UCSF Chimera V1.17. Note that some of the nanobody-related densities appear to have more volume than expected (for example Nb8204); this could be due to severe preferential orientation affecting the reconstruction or the presence of a C-terminal linker and His-tag. All maps and models have been deposited in the EMDB and PDB, respectively. Cryo-EM figures were prepared using ChimeraX V1.1[33]. Cryo-EM data collection and refinement statistics are presented in Supplementary Table 1.

**SEC analysis of nanobody binding to cap-mid-link.** One hundred microliters of cap-mid-link at 1 mg ml⁻¹ was mixed with a 1.5× molar excess of either Nb8193, Nb8194, Nb8195, Nb8197, or Nb8208 and applied to a Superdex 75 Increase 10/300 column equilibrated in 20 mM Na-HEPES pH 7.5, 150 mM NaCl, and 0.5 mM DTT.

**Crystallisation of nanobody-cap-mid-link complexes.** Purified cap-mid-link was mixed with a 1.5× molar excess of either Nb8193, Nb8194, or Nb8208 and applied to a Superdex 75 Increase 10/300 column equilibrated in 20 mM Na-HEPES, pH 7.5, 150 mM NaCl, and 0.5 mM DTT. Fractions containing the cap-mid-link-nanobody complex were concentrated to 10 mg ml⁻¹ and 100 nl was mixed with 100 nl reservoir solution for a sitting drop crystallisation experiment. The Nb8193-cap-mid-link complex formed crystals in a condition containing 20% w/v PEG3350 and 200 mM (NH₄)₂ citrate. The Nb8194-cap-mid-link complex formed crystals in a condition containing 25% w/v PEG3350, 200 mM (NH4)₂SO₄, and 100 mM Tris-HCl, pH 8.5. The Nb8208 complex crystals formed only after 6 weeks in a condition containing 20% w/v PEG3350, 1 M (NH4)₂SO₄, 100 mM BIS-TRIS propane,

pH 5.5. Prior to diffraction experiments all crystals were cryoprotected by addition of 20% v/v glycerol to the crystallisation solution and frozen in liquid nitrogen.

**Diffraction and structure solution of nanobody-cap-mid-link complexes.** All data were collected at Diamond Light Source on beamline I03. Data reduction for all datasets was carried out with autoPROC and STARANISO[40]. Molecular replacement in PHASER[41] was carried out using cap-mid-link (PDB ID 6S5V) and nanobody (PDB ID 6QPG) structures as search models. Molecular replacement for the Nb8193 found two copies of the cap-mid-link construct and two copies of the nanobody in the crystallographic asymmetric unit. For the Nb8194 dataset one copy of the cap-mid-link and one copy of the nanobody was found. Initial searches with complete cap-mid-link failed for the Nb8208 dataset; truncation down to a cap-binding domain only allowed PHASER to place two copies of this domain and two copies of the nanobody. Subsequently iterative rounds of refinement in real and reciprocal space were carried out using COOT[34] and PHENIX[35], respectively. Refinements were carried out without the use of Ramachandran restraints. Data collection and refinement statistics are summarised in Supplementary Table 2. Crystallography figures were prepared with PyMOL.

**In vitro polymerase transcription assay.** In vitro polymerase transcription assays were carried out as previously described[21]. In short, a cap-1 structure was added to a synthetic 11 nucleotides long RNA (5′-ppGAAUACUCAAG-3′) or 20 nucleotides long RNA (5′-ppAAUCUAUAAUAGCAUUAUCC-3′) (ChemGenes) by mixing 1 μM of RNA with 0.25 μM [α-³²P] GTP (3000 Ci mmol⁻¹; Perkin-Elmer), 0.8 mM S-adenosylmethionine, 0.5 U μl⁻¹ vaccinia virus capping enzyme (NEB) and 2.5 U μl⁻¹ 2′-O-methyltransferase (NEB) in a 20 μl reaction at 37 °C for 1 h. The product was isolated using 16% w/v denaturing PAGE, excised, eluted overnight in distilled H₂O and desalted using NAP-10 columns (GE Healthcare). Transcription reactions were performed using the capped RNA primer in a 3 μl reaction mixture containing 0.4 pmoles purified polymerase, 1 mM ATP, 0.5 mM UTP, 0.5 mM CTP, 0.5 mM GTP, 5 mM MgCl₂, 1 mM DTT, 2U ul⁻¹ RNasin, 0.5 μM 5′ vRNA promoter (5′-AGUAGAAACAAGGCC-3′), 0.5 μM 3′ vRNA promoter (5′-GGCCUGCUUUUGCU-3′). Reactions were incubated at 30 °C for 1 h and kept at 0 °C overnight. Equal volume of 2x RNA loading dye containing 80% formamide, 1 mM EDTA and bromophenol blue and xylene cyanol was added followed by incubation at 95 °C for 3 min. Products were resolved by 20% w/v denaturing PAGE and visualised via phosphorimaging on an FLA-5000 scanner (Fuji). Experiments were performed with three independently purified batches of polymerase expressed in insect cells.

**Plasmids.** The pCAGGS-PB1-1918, pCAGGS-PB2-1918, pCAGGS-PA-1918, pCAGGS-NP-1918 plasmids expressing the polymerase subunits and NP of influenza A/Brevig Mission/1/1918 virus have been described[42]. The pcDNA-PB1, pcDNA-PB2, pcDNA-PA, pcDNA-NP plasmids expressing the polymerase subunits and NP of influenza A/NT/60/1968 (H3N2), A/WSN/33 (H1N1) and A/Fujian/01/2002 (H5N1) viruses have been described[43,44]. The plasmid expressing vRNA, encoding a luciferase-reporter gene, under the control of a human RNA polymerase I promoter has also been described[45]. The pcDNA-PB2-1918-TAP plasmid expressing PB2 of influenza A/Brevig Mission/1/1918 with a C-terminal tandem affinity purification (TAP) tag that consists of a calmodulin binding domain (CBD), a tobacco etch virus (TEV) protease cleavage site, and two copies of protein A, was cloned by replacing the PB2 open reading frame of pcDNA-PB2-TAP[46] with that of pCAGGS-PB2-1918. The pHW2000 reverse genetics plasmids for influenza A/WSN/33 have been described[47]. The pHW2000-PB1-1918, pHW2000-PB2-1918, pHW2000-PA-1918 and pHW2000-NP-1918 plasmids were cloned by transferring the open reading frames from the corresponding pCAGGS plasmids and inserting the non-coding 5′ and 3′ terminal sequences of A/Brevig Mission/1/1918[48]. The mammalian cell expression plasmids of C-terminally His-tagged nanobodies were generated by PCR amplification of the of the nanobody genes from the bacterial pMESy4 nanobody expression plasmids[7] and cloning of PCR products into the NcoI and BamHI sites of pcDNA3. All plasmids were confirmed by sequencing.

**Purification of recombinant RNP.** Recombinant RNPs of the 1918 pandemic influenza virus were generated by transfecting 5 μg of each plasmid encoding PB1, PB2, PA and NP of influenza A/Brevig Mission/1/1918 virus and the neuraminidase vRNA of influenza A/WSN/33 virus into ~0.16 × 10⁸ HEK293T cells in a 15 cm dish. Transfections were performed using Lipofectamine 2000 (Invitrogen) according to the manufacturer's instructions. Whole cell lysates were harvested 48 h post transfection. RNPs were affinity-purified using IgG Sepharose chromatography (GE Healthcare) and released using Tobacco Etch Virus (TEV) protease (Thermofisher Scientific) in 2 ml cleavage buffer, containing 10 mM HEPES-NaOH, pH 8.0, 150 mM NaCl, 0.1% v/v Igepal, 1× PMSF, 1 mM DTT and AcTEV (1U per 100 μl IgG Sepharose), as described previously[49].

**In vitro RNP activity assay and primer extension analysis.** A total of 10 μl of recombinant RNP was mixed with 20 ng rabbit globin mRNA (Sigma-Aldrich) as primer donor, 200 ng of purified nanobody or equivalent volume of nanobody purification buffer[7], 1 mM ATP, 0.5 mM GTP, 0.5 mM CTP, 0.5 mM UTP, 5 mM

MgCl$_2$, 1 mM DTT, and 2U μl$^{-1}$ RNasin in a reaction volume of 20 μl. After 4 h incubation at 30 °C, RNA was extracted using TRI reagent (Sigma-Aldrich) according to the manufacturer's instructions. RNA levels were analysed using primer extension as previously described[21]. In brief, RNA was reverse-transcribed using excess $^{32}$P-labelled primers specific to positive-sense mRNA and cRNA. Primer extension products derived from mRNA and cRNA can be distinguished based on size because of the presence of globin mRNA-derived sequences at 5′ end of mRNA. Primer extension products were separated by 6% w/v denaturing PAGE and visualised by phosphorimaging on an FLA-5000 scanner (Fuji). Analysis was carried out using ImageJ (Fiji) and Prism 8 (GraphPad). Experiments were performed with three independently purified batches of RNPs.

**Luciferase minireplicon assay.** Approximately $0.16 \times 10^6$ HEK293T cells were transfected with 0.04 μg of each plasmid encoding PB1, PB2, PA and NP of influenza A/Brevig Mission/1/1918 (H1N1), A/NT/60/1968 (H3N2) or A/Fujian/01/2002 (H5N1) virus, 0.2 μg of plasmid encoding a luciferase vRNA segment, and 0.8 μg nanobody plasmid or an empty pcDNA3 vector. Cells were collected 20 h after transfection, and total cell lysates were obtained by adding 100 μl Luciferase Cell Culture Lysis Reagent (Promega) to the cells, followed by vigorous shaking for 10 min. The presence of luciferase enzyme was determined using a luciferase-based detection system following the manufacturer's instructions (Promega). In brief, 100 μl of Luciferase Assay Reagent (Promega) was added to 20 μl of whole cell lysate in a 96-well opaque plate. The plate was immediately read in microplate reader SpectraMax M3 (Molecular Devices). Two technical replicates were carried out for each transfection and mean values were calculated. These mean values for three independent transfections were used in subsequent analysis in Excel (Microsoft) and Prism 8 (GraphPad).

**Influenza virus reverse genetics.** Recombinant influenza virus was produced by reverse genetics as described previously[47]. In brief, ~$0.6 \times 10^6$ HEK293T cells, cultured in DMEM with 10% FCS, were transfected with 0.5 μg of each pHW2000 plasmid encoding PB1, PB2, PA, NP of A/Brevig Mission/1/1918 and HA, NA, M, NS of A/WSN/33 using Lipofectamine 2000 (Invitrogen) according to the manufacturer's instruction. 24 h post-transfection, cell medium was replaced with DMEM with 0.5% FCS and supernatant was collected 48 h after replacing the medium. The efficiency of reverse genetics was evaluated by titrating the supernatant on MDBK cells, in a standard plaque assay. Viral amplification and serial passages were performed by infecting MDBK cells at an M.O.I. of 0.01 in MEM supplemented with 0.5% FCS and L-glutamine.

**Virus growth analysis.** Approximately $1 \times 10^6$ HEK293T cells were transfected with 5 μg of pcDNA plasmid encoding Nb8190, Nb8191, Nb8192, Nb8207, Nb8208, and Nb8210 or an empty pcDNA3 vector, using Lipofectamine 2000 (Invitrogen) according to the manufacturer's instructions. Subsequently, 20 h after transfection, cells were infected with a reassortant influenza A/WSN/33 virus encoding polymerase and NP genes of A/Brevig Mission/1/1918 at a multiplicity of infection of 0.01. Cell culture medium was collected 12, 24, 36 and 48 h after infection, and virus titres were determined by plaque assay. At each time point, total cellular RNA was also extracted using TRI reagent (Sigma-Aldrich) and viral RNA levels were analysed by primer extension.

**Cell viability assay.** Approximately $1 \times 10^6$ HEK293T cells were transfected with 5 μg of pcDNA plasmid encoding nanobodies or an empty pcDNA3 vector, using Lipofectamine 2000 (Invitrogen) according to the manufacturer's instructions. Cell viability was determined 24 h post-transfection using a CellTiter-Glo luminescent assay (Promega) according to the manufacturer's instructions. Cells infected with a reassortant influenza A/WSN/33 virus encoding polymerase and NP genes of A/Brevig Mission/1/1918 at a multiplicity of infection of 0.01 for 48 h served as positive control.

**Binding assay of influenza virus polymerase to Pol II CTD.** Binding of purified viral polymerase to a four heptad repeat synthetic Pol II CTD mimic peptide phosphorylated on serine-5 of the heptad repeats in the presence of nanobodies was determined as previously described[25]. In brief, 20 μg Pol II CTD mimic peptide were bound to 10 μl streptavidin agarose resin (Thermo Scientific) for 2 h at 4 °C, washed twice with wash buffer (10 mM HEPES [PAA catalogue no. S11-001], 150 mM NaCl, 0.1 % (v/v) Igepal, 1 mM PMSF), blocked with 1% (w/v) bovine serum albumin in wash buffer, before 4 μg of 1918 polymerase, with or without a twofold excess of nanobodies over polymerase, was added to the peptide-coated beads followed by incubation for 1 h at 4 °C. Complexes were washed five times with wash buffer, heated to 95 °C for 5 min in sample buffer and analysed by SDS-PAGE followed by silver staining performed according to the manufacturer's instructions (Invitrogen).

**Primary transcription, cRNA stabilisation and replication assays.** For the primary transcription assay ~$0.2 \times 10^6$ HEK293T cells were transfected with 1 μg of pcDNA nanobody expression plasmids. For the cRNA stabilisation and replication assays the cells were also transfected with 0.4 μg of each of pCAGGS-1918-PB1a

(for cRNA stabilisation assay) or pCAGGS-PB1 (for replication assay) and pCAGGS-PB2, pCAGGS-PA, pCAGGS-NP plasmids. At 48 h post-transfection cells were infected with a reassortant influenza A/WSN/33 virus encoding polymerase and NP genes of A/Brevig Mission/1/1918 at a multiplicity of infection of 5 in the presence of 100 μg/mL cycloheximide. Total cellular RNA was harvested 4 h post-infection using TRI reagent (Sigma-Aldrich) and RNA levels were analysed by primer extension.

**Western blotting.** Western blotting to determine the expression of nanobodies was carried out using a specific goat anti-alpaca polyclonal antibody conjugated to HRP (Jackson ImmunoResearch), while actin was blotted using a rabbit anti-β-actin primary antibody (Sigma-Aldrich) and then goat anti-rabbit secondary antibody conjugated to HRP. Amersham ECL Western Blotting Detection Reagents (GE Healthcare) or Chemiluminescent HRP Substrate (Millipore) were used for detection. Note that the anti-alpaca polyclonal antibody exhibits varying levels of affinity for different nanobodies; hence, band intensities for different nanobodies in western blots are not directly comparable.

**Reporting summary.** Further information on research design is available in the Nature Research Reporting Summary linked to this article.

## Data availability

All data are available from the corresponding authors and/or included in the paper or Supplementary Information. Crystallographic coordinates and maps generated in this study have been deposited in the PDB with accession codes 7NFQ, 7NFR, and 7NFT for the Nb8193, Nb8194, Nb8208 complexes, respectively. Cryo-EM density maps with the corresponding atomic coordinates have been deposited in the Electron Microscopy Data Bank and the Protein Data Bank with the following accession codes for the 1918 polymerase heterotrimer class 1 (PDB 7NHX, EMD-12342), class 2a (PDB 7NHA, EMD-12322), class 2b (PDB 7NHC, EMD-12323), class 3 (PDB 7NI0, EMD-12348). Cryo-EM density maps with the corresponding atomic coordinates have been deposited in the Electron Microscopy Data Bank and the Protein Data Bank with the following accession codes for the complexes between the 1918 polymerase heterotrimer and nanobodies Nb8189 (PDB 7NIK, EMD-12361), Nb8190 (PDB 7NIL, EMD-12362), Nb8191 (PDB 7NIR, EMD-12363), Nb8192 (PDB 7NIS, EMD-12364), Nb8196 (PDB 7NJ3, EMD-12371), Nb8198 (PDB 7NJ4, EMD-12372), Nb8199 (PDB 7NJ5, EMD-12373), Nb8200 (PDB 7NJ7, EMD-12375), Nb8201 (PDB 7NK1, EMD-12428), Nb8202 (PDB 7NK2, EMD-12429), Nb8203 (PDB 7NK4, EMD-12430), Nb8204 (PDB 7NK6, EMD-12431), Nb8205 (PDB 7NK8, EMD-12433), Nb8206 (PDB 7NKA, EMD-12435), Nb8207 (PDB 7NKC, EMD-12437), Nb8209 (PDB 7NKI, EMD-12440), Nb8210 (PDB 7NKR, EMD-12447). Source data are provided with this paper.

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

## Acknowledgements

We thank P. Palese and R. Fouchier for plasmids, I. Berger for the MultiBac system, and G. G. Brownlee, M. Martínez-Alonso, and members of the Fodor and Grimes laboratories for helpful comments and discussions. We also thank Instruct-ERIC, part of the European Strategy Forum on Research Infrastructures (ESFRI), Instruct-ULTRA (EU H2020 Grant 731005), and the Research Foundation—Flanders (FWO) for support with nanobody discovery. We thank Alison Lundqvist for the technical assistance during nanobody discovery. This work was supported by Medical Research Council (MRC) programme grant MR/R009945/1 (to E.F.), Wellcome Investigator Awards 200835/Z/16/Z (to J.M.G.), Clarendon Fund and Medical Science Doctoral Training Centre at the University of Oxford (to Z.Z.), MRC studentships (to A.P.W. and I.S.M.). We thank Diamond Light source for access to the MX beamlines (proposal number MX19946). Electron microscopy provision was provided through the OPIC electron microscopy facility, which was founded by a Wellcome JIF award (060208/Z/00/Z) and is supported by a Wellcome equipment grant (093305/Z/10/Z). Computation was performed at the Oxford Biomedical Research Computing (BMRC) facility, a joint development between the Wellcome Centre for Human Genetics and the Big Data Institute supported by Health Data Research UK and the NIHR Oxford Biomedical Research Centre. Part of this work was supported by Wellcome administrative support grant (203141/Z/16/Z).

## Author contributions

J.R.K., Z.Z., L.C., H.F., E.F and J.M.G. conceived and designed the study. J.R.K., L.C, H.F., I.S.M., carried out cloning of recombinant baculoviruses and protein purification. J.R.K and L.C. collected and processed X-ray crystallography and electron microscopy data and built and refined models. Z.Z., A.P.W. and I.S.M. performed functional assays and analysed data. E.P. and J.S. designed and generated nanobodies. J.M.G. and E.F. supervised the structural and functional studies, respectively. J.R.K., Z.Z., L.C., H.F., E.F. and J.M.G. wrote the paper with input from all co-authors.

## Competing interests

The authors declare no competing interests.
