## [Peer Review File · Nature Communications]

Reviewer #1 (Remarks to the Author):

In this manuscript Keown et al. used a combination of structural characterizations to describe the structure of the A/Brevig/Mission/2/1918 polymerase and determine in an extensive effort the binding sites of 20 nanobodies. The effect of those nanobodies on the influenza polymerase was evaluated in in vitro and in vivo polymerase activity assays, as well as on a subset of nanobodies on viral multiplication. The overall impression of the manuscript is good, the experiments have been carefully performed and the paper is well structured and written. Specific comments can be found below.

General notes:

The study is well conceived and does not overinterpret the findings which are depicted in the data. The nanobodies and their binding sites were described in a rigorous effort and will be of great interest to the community to perform further functional analysis. A limitation of this study is the lack of specific functional evaluations of any of the described nanobodies. The presented data does not allow a precise interpretation of any of the described binding sites and remains therefore elusive.

Specific comments:

- 1) Line 44-46: In the abstract, it's noted that the identified binding sites are likely sensitive to inhibition across influenza A virus subtypes. Without experimental evidence this statement is very speculative, especially for nanobodies which bind the PB2 subunit, which depicts divergence between IAV subtypes.
- 2) Line 87-88: the term "previously unidentified sites of vulnerability" is very vague as this kind of study is new and as stated by the authors the influenza polymerase needs high flexibility, so it's not clear whether most polymerase binding entities would hinder polymerase activity. Moreover, a lot of binding sites have already been connected to a specific function.
- 3) Line 108: The authors used an approach to add the promoter and a capped RNA primer for structural studies. In contrast to other comparable studies (Fan et al., Kouba et al., Wandzik et al.) the capped primer could not be observed in the PB2 cap-binding domain. Could the authors speculate on the reason why this is the case for the A/Brevig/Mission/2/1918 polymerase?
- 4) Line 126: It could be interesting for a lot of readers to comment on PB2 627 in this conformation.
- 5) Line 139 – 162: The nomenclature of this paragraph is hard to follow and not very precise: "The next two structures, In one of the structures, to that in the first structure etc." It could be helpful to find a more precise description of the structures which are compared in this paragraph.
- 6) Line 144: "polymerase active – site?"
- 7) Line 164: The note about ANP32A is very speculative.
- 8) Line 171: As the RNA in the active site is not resolved in this structure, it is very speculative to argue that the RNA template is not the determinant of an ordered priming loop.
- 9) Line 184: The choice of the nomenclature of nanobodies is not obvious to the reader and not easy to follow.
- 10) Line 185: it would be good to be more precise about the production of the presented nanobodies as they are central to this study.
- 11) Line 211: Please indicated where to find the SEC data
- 12) Figure 3b: Changing the representation of minireplicon data to a log scale is more appropriate as replication in this assay is characterized by logarithmic growth. It allows the reader to interpret the window of activity compared to the background and gives a better impression of the potency of inhibition of each nanobody.
- 13) Figure 3: The effect of the nanobodies on the viral polymerase is clear and convincing. However, it would be more convincing if the specificity of these effects as well as effects on cell viability or growth would be investigated.
- 14) Figure 4: Please indicate the subunits of the polymerase

Reviewer #2 (Remarks to the Author):

This manuscript by Keown, Zhu, Carrique and Fan et al describes structures of the 1918 H1N1 influenza strain polymerase complex in three distinct (plus one similar) conformations, which are very similar to previously published influenza polymerase structures. The authors additionally solved the cryoEM structures of 17 polymerase-nanobody complexes to low resolution and co-structures of the C-terminal region together with 3 nanobodies by X-ray crystallography. In total this manuscript contains 24 structures, which is a tremendous amount of work (and sets an insanely high bar for future structural studies). The authors complemented their structural studies with some comparative in vitro and cell-based assays characterising the effect of the different nanobodies on the function of the polymerase. I very much appreciate that the authors provided functional data on different levels of complexity (in vitro, mini-replicon, and infection). The manuscript is well written and the figures are clearly presented. Overall this work is of high quality and I think it will be of interest for the broad readership of Nature Communications. However, there are also a few points that need clarification and some minor issues that require correction.

Points to clarify:

1. What I don't understand is why the authors don't use the H5N1 polymerase as a control in their functional studies as the nanobodies were obtained upon immunisation with this strain's polymerase complex. The comparison and possible conservation of the binding sites between the two polymerases would be interesting to investigate. Instead they kind of limit their functional studies to comparing the effects of the different nanobodies to each other. And although they show the co-structures and functional effects, they don't characterise for example the binding affinity of the nanobody to the polymerase complex. If the aim was solely a proof-of-principle, I would question the novelty here as this has been described previously by the same groups (Fan, H. et al. Structures of influenza A virus RNA polymerase offer insight into viral genome replication. Nature 573, 287–290 (2019)).

2. In the discussion the authors state: "The second site, bound by Nb8208, is located on the cap-binding domain of PB2. This site is occluded in the replicase conformation of the polymerase observed in the dimeric polymerase bound by the host protein ANP32A, suggesting that this nanobody could specifically affect genome replication" It is a bit of a pity that the authors go through all these efforts of mapping the binding sites of the nanobodies to then simply speculate about the possible mechanism when one has to assume that they, as authors of the mentioned ANP32A study, have all the tools at hand to clarify this further. Similar here "Binding at this site would therefore prevent polymerase dimerization and cRNA to vRNA replication specifically" If that was the case, why haven't the authors observed this effect in vitro or on their EM grids?

Minor points:

1. L186 "to a resolution sufficient to accurately position" Please be specific about the resolution range.

2. L 188 "endonuclease resolved in some" Please be specific.

3. L210 "Four of the nanobodies bound to cap-mid-link causing a peak shift in size exclusion chromatography" The corresponding SEC data are not shown and should be included.

4. L212 "Crystallisation of the cap-mid-link-nanobody complexes yielded high resolution structures" please state the resolution. Additionally, datasets of structures 7NFQ and 7NFR shouldn't have been processed to such high resolution for my taste as the statistics of the highest resolution shell are very poor and don't contain much information (7NFQ worse than the other), see $I/\sigma(I)$ and completeness. However, Evans and Murshudov Acta Cryst. 2013 (<https://doi.org/10.1107/S0907444913000061>) demonstrate that including more data, even if weak, does rather help than doing harm, so even though this policy is ok in general the statement "high resolution" is still a bit misleading as the amount of information in that high resolution shells is clearly not very high. I know crystallographers can argue a lot about this point. I therefore just ask the authors to carefully reconsider the resolution cutoff they used and the phrasing in the text.

5. "cap-mid-link construct (residues 247-536 of PB2) from influenza A/duck/Fujian/01/2002

(H5N1)" why was this a different strain than 1918? Maybe the 4th nano body didn't bind because the protein sequence differs quite a bit in the PB2 region?

6. The authors should be precise about the virus strain against which the nanobodies were raised. As it is now, one has to check the two references given and then find out that they were obtained after immunisation with H5N1 virus, the exact strain is even unclear in the reference given (Fan, H. et al. Structures of influenza A virus RNA polymerase offer insight into viral genome replication. Nature 573, 287–290 (2019)).

How could this influence the results of this study? How similar are the proteins (in %)? Are the nanobody binding sites at conserved residues of the pol complex? This should be discussed.

7. I think as a control for the infection experiments the authors performed (Fig 3c) they should test the transient expression of the nanobody over time as this might explain the weak effect of the transient transfection over the course of the infection.

8. Line 290 "as in fact it has been demonstrated for Nb8205" needs a reference to be added

9. Line 294 "These inhibitory sites are highly conserved across the polymerases of influenza A virus strains (Supplementary Fig. 1a-c)" This is difficult to comprehend as the inhibition sites are not labelled in the alignment. A numerical value for identity/similarity would be helpful here.

10. In the discussion the authors describe "five sites" and go from "First site" to "second site" to "the remaining three" again to "the first of these". This is a bit confusing and binding sites could simply be numbered as started in the beginning (1-5).

11. Methods line 558: "100 ng purified polymerase" should be given in molarity.

12. Methods line 598: "Viral RNA levels were analysed using primer extension as previously described. In brief, RNA was reverse-transcribed using 32P-labelled primers specific to positive- and negative-sense viral RNAs. Transcripts were separated by..." I am a bit confused by this description and would appreciate more/precise information here. It is particularly important for the analysis the authors performed to discriminate between positive-strand cRNA and the transcripts. How was this achieved? (This is a manuscript with 24 structures and I don't have loads of time to go through the references in detail. This can be made more digestible for the reader here.)

13. Ext data fig 1: Why is there no RNA marker? The caption says "20 nt long capped RNA requiring prior cleavage by the viral PA endonuclease" but the size of the product RNA is unclear.

14. Ext data fig 4a: It is difficult to see where the lines indicating certain regions end, especially the line indicating residues 624/625. Authors should consider putting arrowheads at the end. I also don't understand why the 90° rotation in the right panel is needed, the respective label is partly occluding the structures.

15. Ext data fig 5: Some of the nanobody-related densities have more volume than expected (e.g. NB8203, NB8192) What could be the reason for this? Structure with NB8204 shows some additional unfilled density at the polymerase core. What would be there?

16. Line 720 "using clustal W." Authors should consider adding the respective reference here.

17. Ext data fig 7c: Why have some nanobodies such as 8207 a higher molecular weight compared to the others?

18. Why does the extended data table 1 not include info on the ramachandran statistics? Were ramachandran or other restraints used in the refinement? This should be indicated in the methods section.

19. For one of the polymerase complex structures the reported resolution of 3.32 Å does not seem to fit to the FSC curves which suggests something around 4.4 Å (PDB report D_1292114034, pages 28+29). Additionally, the number 3.32 does not match with the information provided in the

supplementary table. The same discrepancy is found for the other polymerase structures (PDB report D_1292113981 suggesting 3.45 Å resolution, PDB report D_1292113990 suggesting 3.41 Å resolution, PDB report D_1292114023 suggesting 4.11 Å resolution). All these resolution values are very different from the supplementary table provided and it is thus hard to track which structure belongs to which of the many PDB reports. For the polymerase complex structures the authors should clearly state how the resolution limit was determined (methods section) and check the values provided carefully.

20. For the nanobody co-structures as in PDB report D_1292114098 the half maps haven't been provided, making it impossible for me to judge these data. Overall I had a hard time sorting which report belongs to which structure. This could have really been made easier by the authors.

21. Finally very few grammar issues/typos:

L207 hypothesised

L252 the decreased

L502 "extensive" not needed

L594 excessive "primer"

L697 deepEMhacer should be deepEMhancer

Ext data fig 7b: labelling of NB8191 has a space between 9 and 1

Reviewer #3 (Remarks to the Author):

This work reports structural and functional characterisation of 24 nano bodies directed against the RNA polymerase of 1981 influenza virus.

17 structures of the complex made of one VHH and the polymerase were determined at atomic resolution by cryo-EM and 3 sub-complexes involving other VHH were solved by x-ray crystallography.

All nano bodies were tested using both in vitro and in cellulo functional tests.

This is an impressive work, rather extensive, that could lead to new ideas to inhibit the polymerase by small molecules, potentially leading to new anti-viral compounds.

I only have a few comments for improvement of the manuscript:

1. Figure 1: would it be possible to add a right panel showing the RNA substrate in each case (a, b, c)?

2. line 134 "Alternatively this might be an intermediate position..."

2. I got lost in the numbering and description of the 4 groups (and sub-groups for the fourth group) of the 17 cryo-EM structures (lines 192-205). I suggest to slightly rewrite it using a scaffold along these lines:

-First, ... (5 nano bodies)

-Second,... (3 nano bodies)

-Third,... (1 nano body)

-Fourth, ... (8 nano bodies) in 4 sub-groups: i) 2 VHH, ii) 3 VHH, iii) 2 VHH and iv) 1 VHH

Response to reviewers' comments

Reviewer #1 (Remarks to the Author)

In this manuscript Keown et al. used a combination of structural characterizations to describe the structure of the A/Brevig/Mission/2/1918 polymerase and determine in an extensive effort the binding sites of 20 nanobodies. The effect of those nanobodies on the influenza polymerase was evaluated in in vitro and in vivo polymerase activity assays, as well as on a subset of nanobodies on viral multiplication. The overall impression of the manuscript is good, the experiments have been carefully performed and the paper is well structured and written. Specific comments can be found below.

General notes:

The study is well conceived and does not overinterpret the findings which are depicted in the data. The nanobodies and their binding sites were described in a rigorous effort and will be of great interest to the community to perform further functional analysis. A limitation of this study is the lack of specific functional evaluations of any of the described nanobodies. The presented data does not allow a precise interpretation of any of the described binding sites and remains therefore elusive.

We thank the reviewer for their kind comments. To address the reviewer's general comment, we have added additional functional experiments. These experiments assess the ability of a subset of potent nanobodies to inhibit primary transcription, including Pol II CTD binding, as well as the two steps of genome replication. Using this set of assays we have confirmed many of the suggested modes of action for our nanobodies. See Figure 4a-d.

Specific comments:

1) Line 44-46: In the abstract, it's noted that the identified binding sites are likely sensitive to inhibition across influenza A virus subtypes. Without experimental evidence this statement is very speculative, especially for nanobodies which bind the PB2 subunit, which depicts divergence between IAV subtypes.

We have performed additional luciferase minireplicon assays in the presence of a selection of the nanobodies identified as the most potent in our screening assays, including one that binds to PB2 subunit, across different influenza virus subtypes (H1N1, H3N2, and H5N1). These results show consistent and equivalent inhibition of each of the polymerases from these diverse strains. See Extended Data Fig. 8c.

2) Line 87-88: the term "previously unidentified sites of vulnerability" is very vague as this kind of study is new and as stated by the authors the influenza polymerase needs high flexibility, so it's not clear whether most polymerase binding entities would hinder polymerase activity. Moreover, a lot of binding sites have already been connected to a specific function.

Taking the reviews comments and our new data into consideration we have amended the sentence to now read "Combining functional and structural data we have discovered sites of vulnerability on the polymerase and link these to the inhibition of specific polymerase functions, identifying and validating targets for drug discovery."

3) Line 108: The authors used an approach to add the promotor and a capped RNA primer for structural studies. In contrast to other comparable studies (Fan et al., Kouba et al., Wandzik et

al.) the capped primer could not be observed in the PB2 cap-binding domain. Could the authors speculate on the reason why this is the case for the A/Brevig/Mission/2/1918 polymerase?

We suspect this is related to the dynamics of the flexible polymerase domains, though quantification of these differences is difficult. As the reviewer correctly states the well-ordered “transcription initiation” conformation, with the PB2 C-terminal domains packed closely to the body of the polymerase, has been previously observed with similar capped RNA. These previous studies have used polymerase from human H3N2 or bat H17N10 influenza A viruses or a human influenza B virus. In our conformation the PB2 cap-binding domain is located away from the polymerase body and is therefore less constrained. We speculate that this lack of rigidity makes it difficult to determine whether the capped RNA is present in the cap-binding domain. We now state in the manuscript that we do not fully understand the reasons for the lack of capped RNA in the cap-binding domain.

4) Line 126: It could be interesting for a lot of readers to comment on PB2 627 in this conformation.

The conformation and fold of the 627-domain is identical to that observed in other structures. The position of the 627 residue could not be accurately determined given the relative low resolution of this region of the map. We now state in the manuscript that the fold of the domain is essentially the same as observed in previous structures.

5) Line 139 – 162: The nomenclature of this paragraph is hard to follow and not very precise: “The next two structures, In one of the structures, to that in the first structure etc.” It could be helpful to find a more precise description of the structures which are compared in this paragraph.

We thank the reviewer for their comments. We have amended this section of the text and removed non-specific references to the structures; we now refer to them according to our class 1, class 2a/class2b, and class 3 naming system.

6) Line 144: “polymerase active – site?”

We have added “site” to this sentence.

7) Line 164: The note about ANP32A is very speculative.

We agree that this is an overly strong statement and have amended the language.

8) Line 171: As the RNA in the active site is not resolved in this structure, it is very speculative to argue that the RNA template is not the determinant of an ordered priming loop.

We agree that this sentence is too speculative and have amended the language.

9) Line 184: The choice of the nomenclature of nanobodies is not obvious to the reader and not easy to follow.

This nomenclature is common to all nanobodies produced at the VIB, see Pardon *et al Nature protocols* 2014 and Uchanski *et al Nature Methods* 2021. We have chosen not to rename

these in this work as maintaining this naming system enables us to maintain the lineage of these, especially as two (Nb8205 and Nb8210) have already been published in Fan *et al Nature* 2019.

10) Line 185: it would be good to be more precise about the production of the presented nanobodies as they are central to this study.

We have clarified that this panel of nanobodies was generated against recombinantly expressed purified H5N1 influenza A virus polymerase.

11) Line 211: Please indicated where to find the SEC data

We have included the SEC data showing the binding of the nanobody subset to the cap- mid-link; see Extended Data Fig. 6c.

12) Figure 3b: Changing the representation of minireplicon data to a log scale is more appropriate as replication in this assay is characterized by logarithmic growth. It allows the reader to interpret the window of activity compared to the background and gives a better impression of the potency of inhibition of each nanobody.

We thank and agree with the reviewer that a logarithmic scale would allow better comparison of the inhibition levels amongst the most potent nanobodies. However, it gives an undermining impression of the inhibition levels of the less but still potent nanobodies. We believe that in the screening assays there is a greater need to demonstrate an inhibition even if it is weaker. We would like to also point out that in the influenza virus field minireplicon data are often presented on a linear scale, for example see Long *et al Nature* 2016.

13) Figure 3: The effect of the nanobodies on the viral polymerase is clear and convincing. However, it would be more convincing if the specificity of these effects as well as effects on cell viability or growth would be investigated.

To investigate the specificity of the effects we now use a combination of assays to gain further mechanistic insight into the mode of action of a selected set of nanobodies. These assays address the effect of nanobodies on primary transcription, Pol II CTD binding, as well as the two steps of genome replication, see Figure 4a-d. We have also included data showing that expression of nanobodies has no significant effect on cell viability 24 h post-transfection, see Extended Data Fig. 8b.

14) Figure 4: Please indicate the subunits of the polymerase

We have added the details of the colouring into the figure legends.

Reviewer #2 (Remarks to the Author):

This manuscript by Keown, Zhu, Carrique and Fan et al describes structures of the 1918 H1N1 influenza strain polymerase complex in three distinct (plus one similar) conformations, which are very similar to previously published influenza polymerase structures. The authors additionally solved the cryoEM structures of 17 polymerase-nanobody complexes to low resolution and co-structures of the C-terminal region together with 3 nanobodies by X-ray crystallography. In total this manuscript contains 24 structures, which is a tremendous amount of work (and sets an insanely high bar for future structural studies). The authors complemented their structural studies with some comparative in vitro and cell-based assays characterising the effect of the different nanobodies on the function of the polymerase. I very much appreciate that the authors provided functional data on different levels of complexity (in vitro, mini-replicon, and infection). The manuscript is well written and the figures are clearly presented. Overall this work is of high quality and I think it will be of interest for the broad readership of Nature Communications. However, there are also a few points that need clarification and some minor issues that require correction.

We thank the reviewer for the very positive comments on our manuscript.

Points to clarify:

1. What I don't understand is why the authors don't use the H5N1 polymerase as a control in their functional studies as the nanobodies were obtained upon immunisation with this strain's polymerase complex. The comparison and possible conservation of the binding sites between the two polymerases would be interesting to investigate. Instead they kind of limit their functional studies to comparing the effects of the different nanobodies to each other. And although they show the co-structures and functional effects, they don't characterise for example the binding affinity of the nanobody to the polymerase complex. If the aim was solely a proof-of-principle, I would question the novelty here as this has been described previously by the same groups (Fan, H. et al. Structures of influenza A virus RNA polymerase offer insight into viral genome replication. Nature 573, 287–290 (2019)).

We thank the reviewer for these comments. Comparison of amino acid sequences across the binding sites between the H1N1 H3N2, H5N1 and H17N10 influenza A virus polymerases can be found in Supplementary Fig 1. We have annotated this figure to indicate nanobody binding sites to allow the reader to assess amino acid conservation at particular sites across polymerases from different influenza A virus subtypes. To compare the effect of the inhibitory nanobodies across polymerases from different influenza A virus subtypes we have performed additional luciferase minireplicon assays in the presence of the most potent nanobodies across H1N1, H3N2 and H5N1 influenza A virus polymerases, and shown consistent inhibition across these subtypes, see extended Figure 8c.

2. In the discussion the authors state: “The second site, bound by Nb8208, is located on the cap-binding domain of PB2. This site is occluded in the replicase conformation of the polymerase observed in the dimeric polymerase bound by the host protein ANP32A, suggesting that this nanobody could specifically affect genome replication” It is a bit of a pity that the authors go through all these efforts of mapping the binding sites of the nanobodies to then simply speculate about the possible mechanism when one has to assume that they, as authors of the mentioned ANP32A study, have all the tools at hand to clarify this further. Similar here “Binding at this site would therefore prevent polymerase dimerization and

cRNA to vRNA replication specifically” If that was the case, why haven’t the authors observed this effect in vitro or on their EM grids?

We are grateful for the reviewer’s comments and have conducted further investigations, both *in vitro* and in the context of infection, to address the possible inhibitory mechanisms of the most potent nanobodies, see Figure 4a-d. We now show that Nb8208 significantly reduces cRNA levels and vRNA levels without affecting mRNA levels which is in agreement with the inhibitory mechanism as we have originally proposed.

Minor points:

1. L186 “to a resolution sufficient to accurately position” Please be specific about the resolution range.

We are now specific about the resolution range of our maps.

2. L 188 “endonuclease resolved in some” Please be specific.

We have amended the text to and now state that the PA endonuclease was resolved in a majority of models (excluding Nb8189, Nb8190, Nb8192, Nb8196, Nb8202).

3. L210 “Four of the nanobodies bound to cap-mid-link causing a peak shift in size exclusion chromatography” The corresponding SEC data are not shown and should be included.

We have added in the SEC data showing the binding of the nanobody subset to cap-mid-link, see Extended Data Fig. 6c.

4. L212 “Crystallisation of the cap-mid-link-nanobody complexes yielded high resolution structures” please state the resolution. Additionally, datasets of structures 7NFQ and 7NFR shouldn’t have been processed to such high resolution for my taste as the statistics of the highest resolution shell are very poor and don’t contain much information (7NFQ worse than the other), see $I/\sigma(I)$ and completeness. However, Evans and Murshudov Acta Cryst. 2013 (<https://doi.org/10.1107/S0907444913000061>) demonstrate that including more data, even if weak, does rather help than doing harm, so even though this policy is ok in general the statement “high resolution” is still a bit misleading as the amount of information in that high resolution shells is clearly not very high. I know crystallographers can argue a lot about this point. I therefore just ask the authors to carefully reconsider the resolution cutoff they used and the phrasing in the text.

We agree that the use of “high resolution” here is not appropriate. We have amended the text accordingly to remove references to “high resolution” structures and instead have stated the processed resolution for each of the datasets. We believe that our processing of the data, by applying an anisotropic resolution cutoff, is the correct approach to produce the most correct model. As the reviewer points out, and we agree, the literature is generally clear that incorporation of weak high resolution data can only increase (or at least not worsen) data quality.

5. “cap-mid-link construct (residues 247-536 of PB2) from influenza A/duck/Fujian/01/2002 (H5N1)” why was this a different strain than 1918? Maybe the 4th nano body didn’t bind because the protein sequence differs quite a bit in the PB2 region?

Unfortunately, we were unable to generate crystals for the 1918 version of cap-mid-link. As the H5N1 complete polymerase heterotrimer was used to generate this panel of nanobodies we instead choose to use the H5N1 cap-mid-link construct. As for Nb1985, we can show by SEC and SDS-PAGE that the 1918 polymerase does bind Nb8195; however, we have not been able to determine the binding site by cryo-EM on the complete heterotrimer or crystallography of the cap-mid-link domain.

6. The authors should be precise about the virus strain against which the nanobodies were raised. As it is now, one has to check the two references given and then find out that they were obtained after immunisation with H5N1 virus, the exact strain is even unclear in the reference given (Fan, H. et al. Structures of influenza A virus RNA polymerase offer insight into viral genome replication. *Nature* 573, 287–290 (2019)). How could this influence the results of this study? How similar are the proteins (in %)? Are the nanobody binding sites at conserved residues of the pol complex? This should be discussed.

We now state in the main text of the manuscript that these nanobodies were generated against recombinantly expressed and purified H5N1 influenza A virus polymerase. As shown in Supplementary Fig. 1 the H1N1 and H5N1 polymerases are highly conserved at the amino acid level, with a sequence identity of >97%. We have annotated this figure to indicate nanobody binding sites to allow the reader to assess amino acid conservation at particular sites across polymerases from different influenza A virus subtypes. We have also included additional functional data in the form of a luciferase minigenome assays showing that our subset of potent nanobodies are equally potent against H1N1, H3N2, and H5N1 polymerases.

7. I think as a control for the infection experiments the authors performed (Fig 3c) they should test the transient expression of the nanobody over time as this might explain the weak effect of the transient transfection over the course of the infection.

Influenza virus infection is known to induce host shut-off and production of nanobodies is likely to be suppressed as infection proceeds. Therefore, we agree with the reviewer that lower expression of nanobodies might explain the weaker effects of nanobodies late in infection. To address this we have performed a western blot of nanobodies at the onset of infection (24 h post-transfection) and at 48 h post-infection. For the nanobodies above the detection level, we observed no significant difference of nanobody levels at 24 h post-transfection vs 48 h post-infection. However, the half-life of nanobodies in cells is unknown, hence the effect of host shut-off on nanobody production cannot be determined; it is also not clear whether nanobodies, even if present in cells late in infection, are still available for polymerase binding. Given these uncertainties, we prefer not to discuss this in the manuscript.

8. Line 290 “as in fact it has been demonstrated for Nb8205” needs a reference to be added

We have included a citation to Fan *et al Nature* 2019.

9. Line 294 “These inhibitory sites are highly conserved across the polymerases of influenza A virus strains (Supplementary Fig. 1a-c)” This is difficult to comprehend as the inhibition sites are not labelled in the alignment. A numerical value for identity/similarity would be helpful here.

To help with the understanding of the conservation of different binding sites we have made two changes to the manuscript. We have grouped the nanobodies according to the sites where they bind (site 1, site 2, site 3, sites 4a-d, and site 5a-b). We have incorporated this naming convention into the text and have annotated these in Supplementary Fig. 1 displaying sequence alignments of a set of influenza virus polymerases to allow the reader to assess amino acid conservation at particular sites across polymerases from different influenza A virus subtypes. For the identity/similarity it is difficult to give precise values here as it depends on which strains are being compared. Depending on the exact set of influenza strains we find the polymerase is often >97% conserved.

10. In the discussion the authors describe “five sites” and go from “First site” to “second site” to “the remaining three” again to “the first of these”. This is a bit confusing and binding sites could simply be numbered as started in the beginning (1-5).

We agree with the reviewer and have revised the text; we now use our new naming conventions (site 1, 2, 3, 4a-d, 5a-b) to refer to these sites as we define them earlier in the text.

11. Methods line 558: “100 ng purified polymerase” should be given in molarity.

We have updated this quantity to 0.4 pmoles.

12. Methods line 598: “Viral RNA levels were analysed using primer extension as previously described. In brief, RNA was reverse-transcribed using ³²P-labelled primers specific to positive- and negative-sense viral RNAs. Transcripts were separated by...” I am a bit confused by this description and would appreciate more/precise information here. It is particularly important for the analysis the authors performed to discriminate between positive-strand cRNA and the transcripts. How was this achieved? (This is a manuscript with 24 structures and I don’t have loads of time to go through the references in detail. This can be made more digestible for the reader here).

We apologise for the confusion; we now explain in the methods how positive-sense cRNA and mRNA are distinguished thanks to the presence of a 10-15 nucleotides long additional sequence at the 5’ end of viral mRNAs obtained by cap-snatching.

13. Ext data fig 1: Why is there no RNA marker? The caption says “20 nt long capped RNA requiring prior cleavage by the viral PA endonuclease” but the size of the product RNA is unclear.

We apologise for the confusion. The 11 and 20 nucleotide long capped RNAs, produced synthetically before enzymatic capping, serve as size markers on this gel as indicated.

14. Ext data fig 4a: It is difficult to see where the lines indicating certain regions end, especially the line indicating residues 624/625. Authors should consider putting arrowheads at the end. I also don’t understand why the 90° rotation in the right panel is needed, the respective label is partly occluding the structures.

We have added arrowheads to the lines as suggested. We believe the right hand panel, showing the top-down view down the axis of rotation, clarifies the movement of the domain. We have moved the annotation away from the structural representation. We have kept the

black dot as this is the centre of rotation for the densities as it has been defined in the figure legends.

15. Ext data fig 5: Some of the nanobody-related densities have more volume than expected (e.g. NB8203, NB8192) What could be the reason for this? Structure with NB8204 shows some additional unfilled density at the polymerase core. What would be there?

We observed that a number of the nanobody complex datasets suffered from severe preferential orientation. While these maps were still of sufficient quality to unambiguously position the nanobody it led to some anisotropic resolution distortions in the resulting maps. This is the case for the nanobodies mentioned above. Nb8204 had probably the most severe case of orientation bias; we believe that this bias resulted in the extra noisy density protruding from the nanobody density. Furthermore, the nanobodies have a 20 amino acid residue linker-His-tag at the C-terminus. This will also contribute to extra smeary density around a small nanobody. We have added a note to the methods to address this.

16. Line 720 “using clustal W.” Authors should consider adding the respective reference here.

We have removed the mention of clustal W from the figure legend and instead included a description and reference in the methods section.

17. Ext data fig 7c: Why have some nanobodies such as 8207 a higher molecular weight compared to the others?

Though the nanobodies share a largely conserved scaffold the CDR's (see Extended Data Fig. 6a) are of significantly different lengths causing the differences observed in the SDS-PAGE.

18. Why does the extended data table 1 not include info on the Ramachandran statistics? Were Ramachandran or other restraints used in the refinement? This should be indicated in the methods section.

We have updated Table 2 (the crystallographic table) to now include Ramachandran values and have added a sentence to the methods stating that we did not use Ramachandran restraints in the refinement of the crystal structures. If the reviewer did in fact mean Table 1 (cryo-EM table) Ramachandran values are reported for the 1918 polymerase structure in the absence of nanobodies. We did not report these for the complex structures as these models were only rigid-body fits and the individual components (the polymerase heterotrimer and nanobody) were not refined.

19. For one of the polymerase complex structures the reported resolution of 3.32 Å does not seem to fit to the FSC curves which suggests something around 4.4 Å (PDB report D_1292114034, pages 28+29). Additionally, the number 3.32 does not match with the information provided in the supplementary table. The same discrepancy is found for the other polymerase structures (PDB report D_1292113981 suggesting 3.45 Å resolution, PDB report D_1292113990 suggesting 3.41 Å resolution, PDB report D_1292114023 suggesting 4.11 Å resolution). All these resolution values are very different from the supplementary table provided and it is thus hard to track which structure belongs to which of the many PDB

reports. For the polymerase complex structures the authors should clearly state how the resolution limit was determined (methods section) and check the values provided carefully.

We apologise for not labelling the validation reports more clearly. We have now re-labelled them to match the naming convention used in our manuscript. We believe the difference in the reported resolutions of from PDB validation vs what we report in our manuscript is from the effect of masking. For our resolution determination cyosparc has applied a mask around the protein. The FSC that were supplied to the PDB were generated using the same half maps using the EMDB server (<https://www.ebi.ac.uk/emdb/validation/fsc/>) which does not use a mask in the FSC calculation.

20. For the nanobody co-structures as in PDB report D_1292114098 the half maps haven't been provided, making it impossible for me to judge these data. Overall I had a hard time sorting which report belongs to which structure. This could have really been made easier by the authors.

Each of the reports has now been renamed so the file name matches that of the structure to hopefully make assessment of the work easier.

21. Finally very few grammar issues/typos:

L207 hypothesised **Fixed**

L252 the decreased **Fixed**

L502 "extensive" not needed **Fixed**

L594 excessive "primer" **We have removed the extra "primer"**

L697 deepEMhacer should be deepEMhancer **Fixed**

Ext data fig 7b: labelling of NB8191 has a space between 9 and 1 **Fixed**

Reviewer #3 (Remarks to the Author):

This work reports structural and functional characterisation of 24 nano bodies directed against the RNA polymerase of 1981 influenza virus. 17 structures of the complex made of one VHH and the polymerase were determined at atomic resolution by cryo-EM and 3 sub-complexes involving other VHH were solved by x-ray crystallography. All nano bodies were tested using both in vitro and in cellulo functional tests. This is an impressive work, rather extensive, that could lead to new ideas to inhibit the polymerase by small molecules, potentially leading to new anti-viral compounds.

We thank the reviewer for their kind comments on our study.

I only have a few comments for improvement of the manuscript:
1. Figure 1: would it be possible to add a right panel showing the RNA substrate in each case (a, b, c)?

The RNA used is identical across all these structures and the sequence is specified in the methods. We state in the text that “both 5’ and 3’ promoters are observed in all structures”. While it would be possible to add a panel to indicate the sequence of the RNAs used we think this is not central to the manuscript and could distract from the main message.

2. line 134 "Alternatively this might be an intermediate position..."

We have altered the sentence to read “Alternatively this may be an ...”

2. I got lost in the numbering and description of the 4 groups (and sub-groups for the fourth group) of the 17 cryo-EM structures (lines 192-205). I suggest to slightly rewrite it using a scaffold along these lines: -First, ... (5 nano bodies) -Second,... (3 nano bodies) -Third,... (1 nano body) -Fourth, ... (8 nano bodies) in 4 sub-groups: i) 2 VHH, ii) 3 VHH, iii) 2 VHH and iv) 1 VHH

Thank you for this suggestion. We have accepted the reviewer’s advice and have grouped the nanobodies according to the site where they bind (site 1, site 2, site 3, site 4a-d, and site 5a-b). We have incorporated these into the results section of the text and have annotated these in Supplementary Figure 1 displaying sequence alignments of a set of influenza virus polymerases.

Peer review further comments -

Reviewer #1 (Remarks to the Author):

The reviewers have replied to all my raised points in a sufficient way.

I disagree with one argument, which the authors may consider to change.

12) Figure 3b: Changing the representation of minireplicon data to a log scale is more appropriate as replication in this assay is characterized by logarithmic growth. It allows the reader to interpret the window of activity compared to the background and gives a better impression of the potency of inhibition of each nanobody.

We thank and agree with the reviewer that a logarithmic scale would allow better comparison of the inhibition levels amongst the most potent nanobodies. However, it gives an undermining impression of the inhibition levels of the less but still potent nanobodies. We believe that in the screening assays there is a greater need to demonstrate an inhibition even if it is weaker. We would like to also point out that in the influenza virus field minireplicon data are often presented on a linear scale, for example see Long et al Nature 2016.

I don't think it's a good argument that the influenza field is often presenting it like this as it does not address my point that it is a semi-logarithmic assay. I don't agree that it will undermine the impression of inhibition levels as the data is significant. The added Minireplicon data in supp Figure 8c further illustrates the problem with this representation as the effect on Fujan and 1918 (H1N1) are different but they appear similar in this presentation for the reader and cannot be judged at all for NT60 (H3N2).

Reviewer #2 (Remarks to the Author):

I want to thank the authors for addressing all my comments so carefully. As already said in the initial round of review this work is of high quality and I hope to see it published in Nature Communications soon.

I have found two very minor points the authors might want to look at before final publication:

1. Fig 4b: For all other figures you have provided a P value you consider significant, what about the quantification results in Fig 4b?

2. Extended data figure 6c: please consider including a calibration of the column in the figure so that is obvious for readers which native sizes of the complexes you observe.

Response to Reviewers' comments

Reviewer #1 (Remarks to the Author):

The reviewers have replied to all my raised points in a sufficient way.

I disagree with one argument, which the authors may consider to change.

12) Figure 3b: Changing the representation of minireplicon data to a log scale is more appropriate as replication in this assay is characterized by logarithmic growth. It allows the reader to interpret the window of activity compared to the background and gives a better impression of the potency of inhibition of each nanobody.

We thank and agree with the reviewer that a logarithmic scale would allow better comparison of the inhibition levels amongst the most potent nanobodies. However, it gives an undermining impression of the inhibition levels of the less but still potent nanobodies. We believe that in the screening assays there is a greater need to demonstrate an inhibition even if it is weaker. We would like to also point out that in the influenza virus field minireplicon data are often presented on a linear scale, for example see Long et al Nature 2016.

I don't think it's a good argument that the influenza field is often presenting it like this as it does not address my point that it is a semi-logarithmic assay. I don't agree that it will undermine the impression of inhibition levels as the data is significant. The added Minireplicon data in supp Figure 8c further illustrates the problem with this representation as the effect on Fujan and 1918 (H1N1) are different but they appear similar in this presentation for the reader and cannot be judged at all for NT60 (H3N2).

In response to the reviewer's comment and editor's advice we have changed the figures in question to use a logarithmic scale.

Reviewer #2 (Remarks to the Author):

I want to thank the authors for addressing all my comments so carefully. As already said in the initial round of review this work is of high quality and I hope to see it published in Nature Communications soon.

I have found two very minor points the authors might want to look at before final publication:

1. Fig 4b: For all other figures you have provided a P value you consider significant, what about the quantification results in Fig 4b?

We have now indicated in the figure legend that we consider a p-value of <0.05 to be significant.

2. Extended data figure 6c: please consider including a calibration of the column in the figure so that is obvious for readers which native sizes of the complexes you observe.

We have now indicated in the figure the elution position of molecular weight standards.